# Divorce and adolescent academic achievement: Heterogeneity in the associations by parental education

**Sondre Aasen Nilsen**[1,2]*, **Kyrre Breivik**[1], **Bente Wold**[3], **Kristin Gärtner Askeland**[1], **Børge Sivertsen**[4,5,6], **Mari Hysing**[1,2], **Tormod Bøe**[1,2]

**1** Regional Centre for Child and Youth Mental Health and Child Welfare, NORCE Norwegian Research Centre, Bergen, Norway, **2** Department of Psychosocial Science, Faculty of Psychology, University of Bergen, Bergen, Norway, **3** Department of Health Promotion and Development, Faculty of Psychology, University of Bergen, Bergen, Norway, **4** Department of Health Promotion, Norwegian Institute of Public Health, Bergen, Norway, **5** Department of Research & Innovation, Helse Fonna HF, Haugesund, Norway, **6** Department of Mental Health, Norwegian University of Science and Technology, Trondheim, Norway

* Sondre.Nilsen@norceresearch.no

**Data Availability Statement:** Norwegian Health research legislation and the Norwegian Ethics committees require explicit consent from participants in order to transfer health research

## Abstract

### Background

The link between parental divorce and adolescents' academic achievement may depend on parental educational levels. However, findings have been inconsistent regarding whether the negative associations between parental divorce and adolescents' academic outcomes are greater or smaller in highly educated families. The present study aimed to investigate the possible heterogeneity in the associations between divorce and adolescents' academic achievement by parental educational levels, within the context of the elaborate Norwegian welfare state.

### Methods

The population-based cross-sectional youth@hordaland study of adolescents aged 16–19 years conducted in Norway in 2012, provided information about parental divorce and was linked to national administrative registries ($N$ = 9,166) to obtain high-quality, objective data on the adolescents' grade point average (GPA), and their parents' educational qualifications and income.

### Results

The negative association between parental divorce and GPA was stronger among adolescents with educated or highly educated parents compared to adolescents with less educated parents. This heterogeneity was driven by maternal educational qualifications, whereby divorce was more strongly and negatively associated with GPA among adolescents with educated mothers compared to those with less educated mothers, independent of paternal educational levels and income measures.

data outside of Norway. In this specific case, ethics approval is also contingent on storing the research data on secure storage facilities located in our research institution. Data are from the Norwegian youth@hordaland study whose authors may be contacted at bib@uni.no.

**Funding:** This project has been made possible by the Norwegian ExtraFoundation for Health and Rehabilitation. SAN received funding (Grant number: 2017/FO149543)

**Competing interests:** The authors have declared that no competing interests exist.

## Conclusions

Among adolescents whose parents have low educational qualifications, parental divorce is not associated with their academic achievement. Educated divorced mothers appear less likely to transfer their educational advantages onto their children than nondivorced equally educated mothers, perhaps due to a "double-burden" regarding work pressure and child-rearing responsibilities. There is a need for future studies to detail the mechanisms underlying this finding.

## Introduction

Children and adolescents with divorced or separated parents are less well-adjusted on *average* across a spectrum of outcomes, including physical and mental health, and do less well in school compared to those who grow up with nondivorced parents [1–3]. Adolescents whose parents divorce have been found to experience a decline in overall grade point average (GPA) of one quarter to one-third of a letter grade and to fail more classes than those continuously living with both parents [4]. In general, recent studies suggest that the association between divorce and youths' academic success is partly causal [5,6]. This link between divorce and poorer academic achievement is important, as successful schooling may have a long-term impact on later educational attainment, occupational and economic stability [7], and future physical and mental health [8–10].

To advance research on divorce and youth adjustment, we need to move beyond focusing on averages and try to elucidate for whom and under which circumstances divorce might be associated with adverse outcomes [1]. It is well established that parents' educational attainment is a strong predictor of their children's academic achievement [11,12]. Educated parents monitor their children's academic progress more closely, have more realistic expectations of their academic abilities, and apply more optimal parenting strategies than less educated peers; all factors linked to positive school outcomes among youth [11,13–15]. Further, educated parents may be better able than less educated parents to cope with a divorce due to having greater financial resources and otherwise being more robust against the often-stressful situation that divorce entails [16]. Thus, at face value, it could be reasonable to assume that highly educated parents, on average, buffer their children more against possible negative consequences of divorce on their academic achievement.

Interestingly, two contrasting theoretical perspectives have been proposed in explaining how a divorce might differently impact children's academic achievement by parental educational levels, and both have received empirical support. Building on the above argument, the *compensatory class* hypothesis posits that adverse life events (such as parental divorce or separation) are less harmful to children from higher class (i.e., highly educated) families [17]. Greater financial, social, and parental resources among educated parents might enable them to plan ahead and counteract possible adverse post-separation effects on their children [17,18]. In support of this perspective, a few studies have found that highly educated parents buffer their children against negative consequences of divorce on measures such as math and reading skills, GPA, and later educational attainment [17,19,20].

The *floor effects* hypothesis, on the other hand, states that children in less educated families have limited access to financial, social, and parental resources to begin with. Thus, they have less to lose from a divorce than peers from more affluent families, who may experience a more

substantial reduction in financial and parental resources invested in them [16,18,21]. Several recent studies support this perspective, whereby the educational disadvantages associated with divorce are relatively larger among children with highly educated parents [16,21–24]. For example, Martin [23] found a larger negative association of divorce among children with highly educated compared to children with less educated parents across current math test scores, GPA, and later educational transitions (e.g., high school completion).

Recent efforts have been made in trying to reconcile these contradictory findings (see [21,22]). Firstly, they may stem from differences in measurements and analyses across studies. Importantly, including both parents' educational qualifications in the analyses appears necessary to capture the parental and economic resources available to the child when one of the parents moves out of the home. Relying solely on maternal educational levels, for instance, may not capture the loss in resources experienced when an educated father moves out, and have been found to remove heterogeneity in the outcomes of divorce on adolescents' educational attainment [22]. Moreover, the results might depend on the educational outcome; while findings regarding later educational attainment tend to favor the floor effects hypothesis [16,21–24], studies investigating current school performance (e.g., subject grades, GPA, test scores) have yielded more inconsistent results [17,20,21,23].

The diverging results may also stem from societal factors that vary across countries and time periods, possibly impacting the mechanisms driving heterogeneous outcomes. The economic cost of divorce has traditionally been higher for women than men in several western countries [25,26]. Thus, children with an educated father may lose relatively more financial resources following divorce (as the father often moves out) than peers with a less educated father. Supporting this notion, two studies utilizing data from the 1970 British Cohort Study found that the relatively larger educational consequences of divorce among children with highly educated parents primarily was driven by the lost access to the fathers' resources [21,22]. However, as noted by the authors, only a minority of nonresident fathers paid child support during this period in Britain. Thus, the role of fathers' resources might be less important in other contexts. Indeed, a study on a somewhat more recent US sample found that maternal education was relatively more important than paternal education, whereby highly educated divorced *mothers* were less likely to transfer their educational advantages onto their children [23]. This finding was partly explained through highly educated divorced mothers' relatively lower academic expectations, involvement in school, and leisure activities, compared to nondivorced peers. As put by the author [23]; "*High status single mothers are accomplished, but frequently time constrained*".

Another potential source of diverging results may stem from the commonality of divorce across different educational strata across countries. A recent study found that parental divorce was more detrimental to the educational attainment among children whose parents had a low likelihood of divorce, compared to those with parents with a high likelihood of divorce [24]. A divorce might thus come as more of a shock among families unprepared for disruption, in turn negatively impacting the children from a perhaps otherwise privileged background. Among children with parents at a high risk of divorce, a divorce might rather be one of many adverse events faced during childhood.

Previous studies have primarily been conducted on British, US and German samples on cohorts from the 1970–1980s, and it is uncertain whether previous findings generalize to other contexts. In keeping with the stated need for studies on more recent samples and in other cultural contexts [21,22], this study sought to investigate the potential heterogeneity in the associations between parental divorce and adolescents' academic achievement within the elaborate Norwegian welfare state.

## The Norwegian context

The Norwegian welfare state is amongst the biggest spenders on welfare in the world [27]. It provides an elaborate social safety net through free access to the health care system and access to sickness, unemployment, and family-related benefits. Levels of absolute deprivation and income inequality in Norway is low [28,29], and the population is highly educated; 38.2% of all women and 30.1% of all men had completed some form of university-level education in 2019 [30]. Like in the other Nordic welfare states, the "dual-earner family" is strongly encouraged; public childcare and schools are highly subsidized, and generous parental leave rights (also for the father) have facilitated the combination of full-time employment and childcare among both mothers and fathers (see [31]). Perhaps, as a result, fathers' housework and childcare time are generally high among most groups of fathers, although women still do more than their male partners [32].

The crude divorce rate in Norway has nevertheless more than doubled since the 1970s, and there were approximately two divorces per 1000 persons in 2016 [33]. The risk of divorce is higher for the less educated, whereby a couple where both have low levels of education run a risk that is more than four-fold in magnitude compared to couples where both have higher educational qualifications [34]. After a divorce, custodians are supported by tax deductions, law-regulated cash allowances, and child support, which is enforced by the public authorities. It is estimated that 50% of parents experience a drop in household income following a divorce in Norway [35]. Nevertheless, the Norwegian welfare state appears fairly successful in equalizing gender differences in the cost of divorce, whereby men and women experience an approximately equal 20% decline in disposable income [36]. This contrasts findings from many other countries where women often lose substantially more than men [25,26]. However, a divorce increases the sickness absence among women with children in Norway but less so for men [37]. This is in general keeping with the "double-burden" hypothesis [38], suggesting that high labor participation, coupled with high child-rearing responsibilities, could be an extra burden for divorced, educated women in Norway. Indeed, although the rates of families who share custody have risen in Norway in the last decades, approximately 65% still live in mother custody following divorce [39].

The link between parental divorce and academic achievement is well documented in Norway; divorce has been associated with having more problems in school [40], lower GPA [41,42], and lower probability of completing higher secondary education [43]. The rates of students not completing high-school are higher in Norway than in many other comparable countries [44]. Not receiving a high-school diploma is associated with a higher risk of later receiving medical and non-medical social insurance benefits [45], and thus represents a significant public health concern. Increased knowledge of the links between parental divorce, parental education, and adolescents' school performance, could thus provide insights that can be utilized in efforts aimed at facilitating high-school completion among adolescents.

## The present study

The main aims of the present study were twofold: Firstly, to investigate the association between parental divorce and adolescents' GPA. Secondly, to examine whether parental educational qualifications moderated this association. To aid these aims, we draw on high-quality register-based measures of adolescents' GPA, parental educational qualifications, and household income, that were merged with a population-based study.

The present study contributes to this field by studying heterogeneous associations of divorce in a society that combines generous social benefits; a highly educated population; high levels of labor-force participation among women; gender equity in the cost of divorce; and

high divorce rates and gender differences in sickness absence and health outcomes among divorced individuals.

The high divorce rate among the relatively few uneducated parents in Norway suggests that a divorce perhaps is one of many adverse events experienced in these selected and often socio-economically disadvantaged families. This led us to expect that the negative association between divorce and adolescents' GPA would be stronger in more highly educated families where a divorce might carry greater changes to children's lives. However, as the Norwegian welfare state appears successful in equalizing the cost of divorce between men and women, we suspected that loss of fathers' financial resources following divorce might be less important in understanding potential heterogeneous outcomes in Norway compared to previous studies conducted in other sociopolitical contexts.

## Materials and methods

### Procedure

We used data from the youth@hordaland study, a population-based survey of adolescents aged 16–19, conducted in the spring of 2012 in Hordaland County, Norway. The youth@hordaland study aimed to assess mental health, family life, lifestyle, school performance, and health service use in adolescents. The adolescents received information about the study per e-mail, and one regular school hour was allocated to complete the questionnaire. Those not attending school on the day of the study could complete the questionnaire at their convenience, and some schools arranged catch-up days. A teacher organized the data collection and protected confidentiality. The adolescents themselves indicated if they consented to complete the entire survey or selected parts of it, as Norwegian regulations dictate that individuals aged 16 years and older are required to consent themselves. Their parents were informed about the study, and the study was approved by the Regional Committee for Medical and Health Research Ethics in Western Norway.

### Sample

All adolescents born between 1993 and 1995 and residing in Hordaland at the time of the survey were invited (N = 19,439) to participate, and 10,257 agreed, yielding a participation rate of 53% for the entire study. The present paper is based on a subsample of 9,166 adolescents (47% of the invited population) who consented to register linkage. This subsample was nearly identical to the total sample with regards to age and gender distribution, and self-reported sociodemographics (S1 Table).

### Measurements from registers

**Age and gender.**   Date of birth and gender were obtained through the adolescents' identity number in the Norwegian National Registry. Exact age was calculated from the date of participation in the youth@hordaland study and the birthdate of participants.

**GPA.**   The adolescents' GPA for each year in upper secondary education were obtained from the National education database in Norway (NUBD) that is owned and administered by Statistics Norway. NUBD contains educational statistics from elementary school through PhD-level. In Norway, each subject is graded on a scale ranging from 1 (failure) to 6 (excellent), and the GPA is thus calculated by taking the sum of all grades received in a given school year divided by the total number of subjects. The grades used in the current study stem from the school-year of 2011–2012. Thus, the grades correspond to the school year that the adolescents were in at the time of the youth@hordaland study. A previous publication found that the

mean GPA in the current sample was quite similar to both regional and national statistics, indicating representativeness of the sample [46].

**Parental education.**  The highest completed educational level of both parents when the adolescents were 16 years old were also obtained from NUBD. The International Standard Classification of Education (ISCED) 2011 coding-scheme was utilized to create three main measures of parental educational levels: 1: A combined measure of parents' educational level indicating the highest completed education in the family by either the mother or the father. The categories were (1) both parents have no qualifications higher than lower secondary education (ISCED 0–2), (2) at least one parent has qualifications equivalent to ISCED 3–5 (upper secondary education, post-secondary non-tertiary education, short-cycle tertiary education), at least one parent has education on Bachelor's level or equivalent (ISCED 6), and at least one parent has attained a Master's or Doctoral level of education (ISCED 7–8). This variable aimed at capturing the range of parental educational levels within a manageable and meaningful set of categories.

2: A combined measure of parents' educational level used to investigate the relative importance of maternal and paternal educational levels. The categories were (1) both parents have no qualifications (ISCED 0–2), (2) only the mother has some qualifications (above ISCED 2), (3) only the father has some qualifications (above ISCED 2), and both have some qualifications (above ISCED 2). This operationalization or similar has been utilized by previous studies [21,22], and we report the results of this categorization in order to facilitate comparison with the pre-existing literature.

3: We also created separate variables for maternal and paternal educational levels to investigate how sensitive the estimates were based on the choice creating combined measures of parents' education, and to further detail the relative contribution of maternal and paternal educational qualifications within a broader range of educational levels (i.e., ISCED 0–2, ISCED 3–5, ISCED 6, ISCED 7–8).

**Measures of family finances.**  The Norwegian national income registry provided information on family finances. The Norwegian Government utilizes this data to estimate taxation, and it can be considered to be of high quality. We utilized three measures of income as covariates in the analysis: Mother's and father's net income (i.e., the sum of wages and salaries, income from self-employment, property income and transfers received minus total assessed taxes and negative transfers), and the equivalized disposable income (EDI) in the household occupied by the adolescents. EDI is a measure of income in a household that is adjusted by an equivalence scale. EDI has been documented in prior publications from the youth@hordaland study [47,48]. The current study utilizes the OECD modified scale, which gives the first adult in the household a weight of 1, subsequent adults are given a weight of 0.5, and each child below 14 years of age is given a weight of 0.3 [49]. The equivalence scale thus enables comparison between households of different sizes and compositions. All income measures stem from the year 2011.

## Measures from the youth@hordaland study

**Parental divorce or separation.**  We coded experience of divorce or separation according to the adolescents' answers to the following questions: "Do your biological parents live together?" and "Have your biological parents divorced or separated?". Adolescents stating that their biological parents did not live together and that their biological parents had divorced or separated were categorized as having divorced parents, while those stating that their biological parents still lived together were defined as living in a nondivorced two-parent (i.e., nuclear) family. These items allowed us to separate between adolescents whose parents split apart, from

adolescents whose parents never lived together, were separated due to death, illness or other reasons (which were removed from the analyses), and resulted in a dummy coded variable (0 = nondivorced/nuclear family, 1 = divorced family). The adolescents also reported year of parental divorce or separation allowing us to calculate a variable of years since the event of dissolution.

We had no means of determining whether the parents were legally married. Official statistics report that 73.5% of children and youth below the age of 18 in a two-parent household in Hordaland county in 2012 lived with married parents (the rest with cohabiting parents) [50]. Thus, the nondivorced group in the present study most likely contained a group of adolescents with parents that had cohabitated since their birth. As some cohabiting unions eventually marry, we find it likely that the proportion of cohabiting unions in the present sample was somewhat lower than regional estimates also including younger children. Similarly, the divorced group likely contained a group of adolescents whose parents split up from cohabitation. Unfortunately, no official statistics regarding dissolution from cohabiting unions in Norway exists. Our inability to exactly detail the adolescents' family structure is not unique to the present study but has been rather common within this research field [2]. For ease of exposition, while keeping the aforementioned statistics in mind, we use the term *divorce* to refer to the dissolution of either cohabitating or marital unions.

## Statistical analyses

Ordinary Least Squares (OLS) regression analyses were conducted to investigate the associations between parental divorce, parental education, and the adolescents' GPA. In the first OLS models, we used the highest completed education in the family as a measure of parental education. The regression models were structured as follows: A baseline model estimating the associations between parental divorce and adolescents' GPA, adjusted by gender and age; Model 1 included the measure of parents' highest completed education; Model 2 added the interaction term between education and parental divorce to investigate the possible heterogeneity in the effects of divorce on the adolescents' GPA; and Model 3 further included the equivalized disposable income in the household currently occupied by the adolescent, mother's net income, and father's net income. These income measures would thus shed light on the possible attenuating effects of both maternal and paternal income levels on the associations between divorce, parental educational levels, and their interactions.

Age and all income measures were centered on their respective means in the regression analyses to ease the interpretation of the regression coefficients. The income measures were divided by a factor of 100,000. Thus, the regression coefficients of the income measures indicate the predicted change in the adolescents' GPA by an increase of 100,000 NOK above the mean.

To replicate the categorization of parental education used by several prior studies [21–23], the above models were re-run utilizing the second measure of parental education, separating families where either none, only the mother, only the father, or both parents had educational qualifications greater than ISCED 2.

Lastly, to test the sensitivity of the above models and to investigate further possible differential associations of maternal and paternal educational levels, the analyses were re-run with maternal and paternal education entered as separate predictors, while retaining the full range of educational levels.

In all regression models, the reference categories for the parental educational variables were set at the lowest parental educational level (i.e., ISCED 0–2). Checks were made for other

differences between the educational levels in the association between divorce and the adolescents' GPA by alternating the reference categories.

Incomplete responses were fairly low in the current sample, where the majority of missing values pertained to the divorce status variable (8.8%), followed by father's net income (4.6%) and parental education (3.7%), whereas the remaining variables utilized in the current study had below 3% missingness. Due to the relatively low proportion of incomplete responses, missing values were handled by listwise deletion in the regression analysis.

**Robustness and sensitivity analyses.** Conditioning on measures of income and paternal and maternal education simultaneously may introduce overcontrol bias [51]. In the two first set of regression analyses, we try to avoid this problem by creating single measures combining information on parental education from both parents, and by entering income variables in the last set of models (as we were not interested in the main effects of income per se). In the last set of regressions, we have made robustness checks by entering maternal and paternal education in separate models.

The timing of divorce could potentially covary with parental educational levels and the adolescents' GPA (e.g., if highly educated parents divorced later on, the estimates of divorce by parental education on the adolescents' GPA could be influenced by the proximity to the event of dissolution). Moreover, association between timing of divorce and the adolescents' GPA may depend on parental educational qualifications (i.e., that more time spent with highly educated divorced parents differ from time spent with lowly educated divorced parents). We investigate these issues by comparing years since divorce across parental educational levels, and by graphically plotting potential linear and non-linear relationships between timing of divorce and GPA by parental educational qualifications. Generalized additive models (GAMs) were used to investigate potential non-linear relationships. In brief, GAMs may be considered as a semi-parametric extension of the generalized linear model, with the strength of the ability to detect non-linear structures in data that otherwise might be missed [52].

Lastly, we performed checks utilizing the income measures as alternative indicators of the family's socioeconomic resources. The income measures were divided into quartiles (i.e., into four equal parts representing the lowest 25% to the highest 25%), and the adolescents' GPA was regressed on the interaction term between parental divorce and the income quartiles (similarly to the procedure described above).

All statistical analyses were conducted using R version 3.5.2 for Mac [53]. Figures were created with the packages "ggplot2" [54], "sjPlot" [55], and "ggstatsplot" [56]. The GAMs were plotted with aid from the "mgcv" package [57] within the "geom_smooth" function of the ggplot2 package. For brevity, statistical parameters are included in figures displaying pairwise comparisons. In Table 1, the effect sizes for categorical variables were calculated from the Mahalanobis distance method and compared between groups [58].

## Results

### Characteristics of the sample

There were fewer boys in the divorced sample (43.6%) compared to the nondivorced sample (47.2%). Parents who divorced had lower education; almost twice as many divorced parents did not have higher than ISCED 2 qualifications (6.6%) compared to nondivorced parents (3.8%), and having qualifications equivalent to Bachelor's level (ISCED 6) or Master's or PhD-levels (ISCED 7–8) were more frequent among nondivorced parents compared to divorced parents. Divorced households had lower equivalized disposable income compared to nondivorced households. While nondivorced fathers had higher net earnings than their divorced

Table 1. Sociodemographic statistics of the sample (*n* = 8360).

| | Nondivorced families (*n* = 5809) | Divorced families (*n* = 2551) | Eff.size |
|---|---|---|---|
| | n (%) | n (%) | |
| Age [mean (sd)] | 17.41 (0.84) | 17.42 (0.83) | 0.016 |
| Male | 2744 (47.2) | 1113 (43.6) | 0.072 |
| Years since divorce [mean (sd)] | - | 10.58 (5.20) | |
| Highest completed education in the family | | | 0.267** |
| ISCED 0–2 | 217 (3.8) | 158 (6.6) | |
| ISCED 3–5 | 2273 (39.9) | 1168 (48.8) | |
| ISCED 6 | 2192 (38.4) | 794 (33.2) | |
| ISCED 7–8 | 1019 (17.9) | 271 (11.3) | |
| Parental education above/below ISCED 0–2 | | | 0.284** |
| No parent > ISCED 2 | 217 (3.8) | 158 (6.6) | |
| Only father > ISCED 2 | 610 (10.7) | 320 (13.4) | |
| Only mother > ISCED 2 | 495 (8.7) | 367 (15.3) | |
| Both parents > ISCED 2 | 4379 (76.8) | 1546 (64.7) | |
| Maternal education | | | 0.213** |
| ISCED 0–2 | 848 (14.7) | 512 (20.5) | |
| ISCED 3–5 | 2138 (37.2) | 1027 (41.0) | |
| ISCED 6 | 2281 (39.7) | 816 (32.6) | |
| ISCED 7–8 | 484 (8.4) | 148 (5.9) | |
| Paternal education | | | 0.338** |
| ISCED 0–2 | 722 (12.6) | 531 (22.0) | |
| ISCED 3–5 | 2463 (43.0) | 1150 (47.6) | |
| ISCED 6 | 1755 (30.6) | 537 (22.2) | |
| ISCED 7–8 | 792 (13.8) | 198 (8.2) | |
| Household income measures, in 100, 000 NOK [mean (sd)] | | | |
| Equivalized disposable income | 3.71 (2.42) | 3.02 (1.70) | 0.330** |
| Net income mother | 3.18 (2.05) | 3.59 (2.39) | 0.186** |
| Net income father | 4.89 (4.47) | 4.19 (2.77) | 0.187** |
| Grade point average [mean (sd)] | 4.07 (0.86) | 3.76 (0.94) | 0.339** |

Eff. Size = effect size, as represented by the standardized mean difference. For categorical variables, the effect sizes were calculated from the Mahalanobis distance method. NOK = Norwegian krone.

** $p < 0.01$; p-values derived from chi square tests for categorical variables, and Welch two-sample *t* test for continuous variables.

counterparts, divorced mothers had higher net earnings than nondivorced mothers (see Table 1 for details).

## Regression results

**Highest education in the family.** The first tested OLS models utilizing the highest completed education in the family as a measure of parental education are displayed in Table 2. The baseline model indicated that adolescents with divorced parents on average had 0.30 points lower GPA score (Cohen's *d* = 0.34) compared to their peers with nondivorced parents. Statistically controlling for the highest completed parental educational level (Model 1), reduced the strength of the association between parental divorce and GPA by 0.06 GPA points (20%), indicating that the association between parental divorce and GPA were relatively robust to adjustments for parental educational levels. Independent of parental divorce, higher education in the

**Table 2. Regression estimates of GPA by parental divorce and the highest parental education in the family (*n* = 7,739).**

| | Baseline model | Model 1 | Model 2 | Model 3 |
|---|---|---|---|---|
| | *b* (S.E) | *b* (S.E) | *b* (S.E) | *b* (S.E) |
| Parental divorce (ref. nondivorced) | -0.300 (0.022) ** | -0.240 (0.021) ** | -0.037 (0.092) | -0.048 (0.092) |
| Gender (ref. girl) | -0.207 (0.020) ** | -0.217 (0.019) ** | -0.217 (0.019) ** | -0.218 (0.019) ** |
| Age | -0.087 (0.012) ** | -0.088 (0.011) ** | -0.088 (0.011) ** | -0.089 (0.011) ** |
| Highest education in the family (ref. both ISCED 0–2) | | | | |
| ISCED 3–5 | - | 0.376 (0.048) ** | 0.458 (0.062) ** | 0.448 (0.062) ** |
| ISCED 6 | - | 0.654 (0.048) ** | 0.748 (0.062) ** | 0.728 (0.062) ** |
| ISCED 7–8 | - | 0.921 (0.051) ** | 0.998 (0.065) ** | 0.964 (0.065) ** |
| Highest education x Parental divorce | | | | |
| ISCED 3–5 x Parental divorce | - | - | -0.202 (0.097) * | -0.201 (0.097) * |
| ISCED 6 x Parental divorce | - | - | -0.245 (0.099) * | -0.246 (0.099) * |
| ISCED 7–8 x Parental divorce | - | - | -0.169 (0.109) | -0.172 (0.109) |
| Income measures | | | | |
| Household EDI | - | - | - | -0.011 (0.009) |
| Net income father | - | - | - | 0.010 (0.009) * |
| Net income mother | - | - | - | 0.017 (0.009) ** |
| Constant | 4.181 (0.015) ** | 3.618 (0.047) ** | 3.535 (0.060) ** | 3.556 (0.060) ** |
| Adjusted $R^2$ | 0.042 | 0.108 | 0.108 | 0.110 |

*b* = unstandardized regression coefficient, S.E = standard error, ref. = reference group.

Age and income variables are centered on their respective means. All income measures are presented in 100,000 NOK.

EDI = Equivalized disposable income.

\* $p < 0.05$

\*\* $p < 0.01$

family was associated with a higher GPA. The interactions between divorce and parental education were further added in model 2, while income measures were added in model 3. Taken together, the results from these models showed that the associations between having divorced parents and the adolescents' GPA were significantly stronger among adolescents where the highest parental education was at secondary school levels (ISCED 3–5) or Bachelor's levels (ISCED 6), compared to those with parents that did not have higher than basic-level education (ISCED 0–2). Although the same trend was observed among adolescents with at least one parent with a Master's or PhD-level education (ISCED 7–8), the interaction term was not significant. Including the income measures (Model 3) hardly changed these estimates. The interactions are visually depicted in Fig 1A.

**Maternal and paternal education.** The OLS models with the parental education measure differentiating between families where either mother, father, or both had above ISCED 2 qualifications are displayed in Table 3. The main findings from these models were that the associated reduction in GPA by having divorced parents was significantly larger if only the mother or both parents had above ISCED 2 qualifications, compared to if no parent had above ISCED 2 qualifications. If only the father had above ISCED 2 qualifications, however, no significant interaction with parental divorce was observed (see Fig 1B).

To check whether the estimates from the above models were sensitive to the choice of combining the maternal and paternal educational levels into overall measures parental education, the analyses were re-run by entering paternal and maternal education as two separate and independent variables. These models revealed that the heterogeneity in the associations between divorce and GPA by parental education were driven by maternal educational levels;

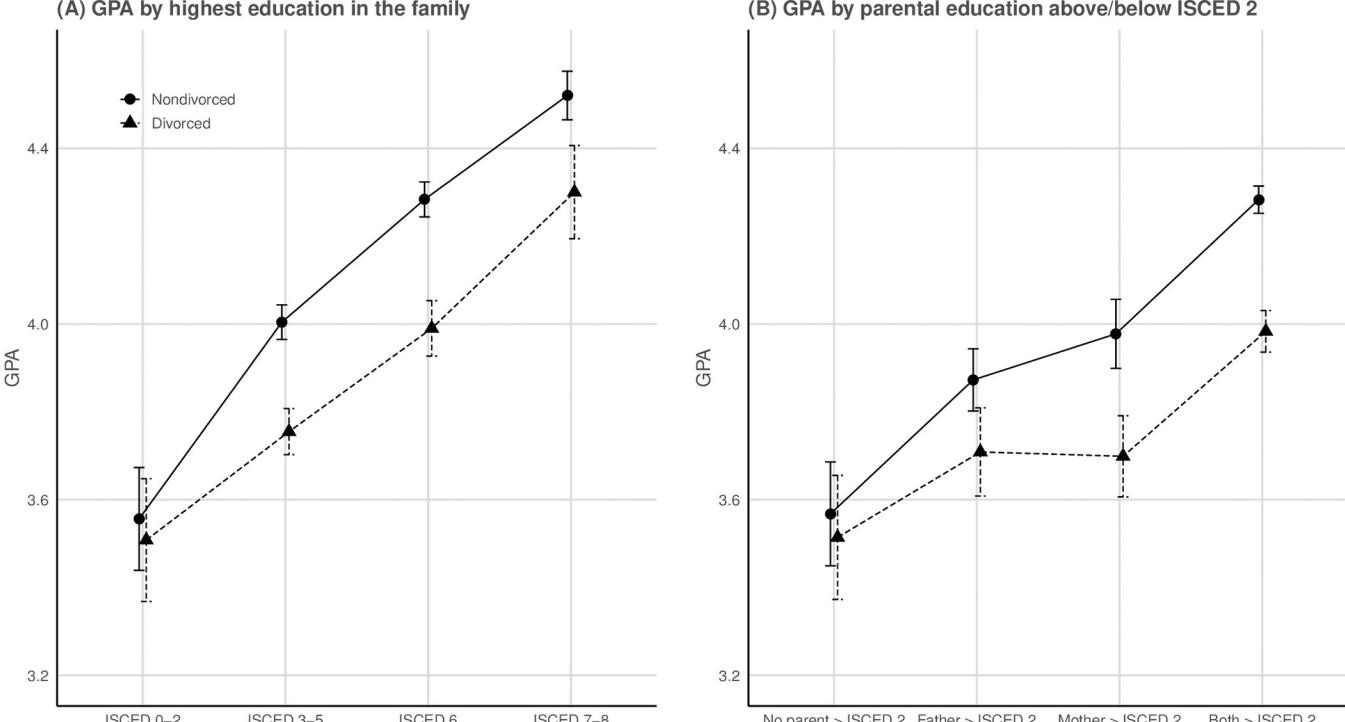

**Fig 1. Predicted values of GPA by parental divorce and highest education in the family (*n* = 7,739).** The predicted values of GPA by (**A**) the highest educational qualifications obtained in the family, and (**B**) by the highest maternal and paternal education above/below ISCED 2 from the fully adjusted regression models (cf. Tables 2 and 3), stratified by divorce status. Error bars represent 95% confidence intervals of *b*. **A**: ISCED 0–2 = up to lower secondary education, ISCED 3–5 = upper secondary education, post-secondary non-tertiary education, short-cycle tertiary education, ISCED 6 = Bachelor's level, ISCED 7–8 = Master's or Doctoral level. **B**: No parent > ISCED 2 = No parent with higher than lower secondary education, Father > ISCED 2 = Only father has above lower secondary education, Mother > ISCED 2 = Only mother has above lower secondary education, Both > ISCED 2 = Both parents have above lower secondary education.

the estimated reduction in GPA by having divorced parents was statistically significantly higher when maternal educational levels were at secondary school levels (ISCED 3–5; *b* = -0.120, *p* < 0.05), at Bachelor's levels (ISCED 6; *b* = -0.175, p < 0.05) and at Master's or PhD-level (ISCED 7–8; *b* = -0.209, *p* < 0.05) compared to basic-level education (ISCED 0–2), after adjustments for paternal education and income measures. There were, however, no significant interaction effects between paternal educational levels and divorce on the adolescents' GPA while holding the effects of maternal education and income constant (see Table 4; Fig 2A and 2B).

Alternating the reference categories of the parental education variables in the regression analyses did not reveal any further statistically significant differences in the links between divorce and GPA by parental educational qualifications (i.e., the main differences were between the ISCED 0–2 levels and the other ISCED levels).

**Robustness checks.**   Entering maternal and paternal educational levels in separate models, in order to check for overcontrol bias, yielded approximately identical estimates. The only exception was that the difference in the relationship between divorce and GPA was slightly smaller and not statistically significantly different (at *p < 0.05*) between the highest maternal educational levels (ISCED 7–8) compared to the lowest maternal educational levels (ISCED 0–2) in the interaction analyses (*b* = -0.163, *p* = 0.096).

Adolescents with highly educated parents experienced, on average, that their parents divorced somewhat later (see Fig 3A and 3B). The mean difference in years since divorce

**Table 3. Regression estimates of GPA by parental divorce and maternal and paternal education above/below ISCED 2 ($n$ = 7,739).**

| | Baseline model | Model 1 | Model 2 | Model 3 |
|---|---|---|---|---|
| | $b$ (S.E) | $b$ (S.E) | $b$ (S.E) | $b$ (S.E) |
| Parental divorce (ref. nondivorced) | -0.300 (0.022) ** | -0.254 (0.021) ** | -0.037 (0.093) | -0.054 (0.093) |
| Gender (ref. girl) | -0.207 (0.020) ** | -0.213 (0.019) ** | -0.213 (0.019) ** | -0.215 (0.019) ** |
| Age | -0.087 (0.012) ** | -0.085 (0.012) ** | -0.085 (0.012) ** | -0.087 (0.012) ** |
| Parental education (ref. both ISCED 0–2) | | | | |
| Only father > ISCED 2 | - | 0.263 (0.054) ** | 0.318 (0.069) ** | 0.305 (0.069) ** |
| Only mother > ISCED 2 | - | 0.335 (0.055) ** | 0.432 (0.071) ** | 0.410 (0.071) ** |
| Both parents > ISCED 2 | - | 0.654 (0.047) ** | 0.751 (0.061) ** | 0.715 (0.061) ** |
| Parental education x Parental divorce | | | | |
| Only father > ISCED 2 x Parental divorce | - | - | -0.111 (0.112) | -0.111 (0.111) |
| Only mother > ISCED 2 x Parental divorce | - | - | -0.236 (0.111) * | -0.225 (0.111)* |
| Both parents > ISCED 2 x Parental divorce | - | - | -0.248 (0.097) * | -0.246 (0.096) * |
| Income measures | | | | |
| Household EDI | - | - | - | -0.014 (0.009) |
| Net income father | - | - | - | 0.015 (0.004) ** |
| Net income mother | - | - | - | 0.027 (0.006) ** |
| Constant | 4.181 (0.015) ** | 3.622 (0.048) ** | 3.533 (0.060) ** | 3.567 (0.060) ** |
| Adjusted R$^2$ | 0.042 | 0.086 | 0.087 | 0.092 |

$b$ = unstandardized regression coefficient, S.E = standard error, ref. = reference group.

Age and income variables are centered on their respective means. All income measures are presented in 100,000 NOK.

EDI = Equivalized disposable income.

* $p < 0.05$

** $p < 0.01$

among highly educated (i.e., ISCED 7–8) vs. lowly educated (i.e., ISCED 0–2) mothers was about 2.2 years, while the comparable figure among fathers was 2.8 years. Plotting the adolescents' GPA as a function of years since divorce across the parental educational qualifications (see Fig 4) revealed a slight negative linear association between years since divorce and GPA across most of both maternal and paternal educational levels. The negative association between time since divorce and GPA was strongest among lowly educated mothers. As lowly educated mothers on average had most years since divorce, this finding highlights that time since divorce could not explain the heterogeneity in the associations between divorce and the adolescents' GPA by maternal educational qualifications. Indeed, the plot suggests that holding years since divorce constant across maternal educational qualifications would slightly strengthen the difference in the negative association between divorce and GPA among adolescents with highly- compared to lowly educated mothers.

The plotted GAM curves show some variability around the linear functions for some of the parental educational levels. Overall, these trends do not give any strong indications that GPA is highly influenced by the timing of divorce in the present study.

Lastly, using equivalized disposable income (EDI) as an alternative indicator of socioeconomic resources, we found a similar but weaker pattern whereby the negative association between divorce and GPA was relatively stronger among adolescents in the second income quartile ($Q_2$; $b$ = - 0.16, $p < 0.01$) and in the fourth quartile ($Q_4$; $b$ = -0.15, $p = 0.02$) compared to those in the first quartile ($Q_1$). The difference between $Q_1$ and $Q_3$ was not statistically significant (see S1 Fig with further test statistics). Adjusting the analyses for parental educational qualifications attenuated and removed the significant difference between $Q_1$ and $Q_4$. No

**Table 4. Regression estimates of GPA by maternal and paternal education ($n$ = 7,739).**

|  | Baseline model | Model 1 | Model 2 | Model 3 |
|---|---|---|---|---|
|  | $b$ (S.E) | $b$ (S.E) | $b$ (S.E) | $b$ (S.E) |
| Parental divorce (ref. nondivorced) | -0.300 (0.022) ** | -0.222 (0.021) ** | -0.099 (0.064) | -0.104 (0.064) |
| Gender (ref. Girl) | -0.207 (0.020) ** | -0.216 (0.019) ** | -0.215 (0.019) ** | -0.216 (0.019) ** |
| Age | -0.087 (0.012) ** | -0.087 (0.011) ** | -0.087 (0.011) ** | -0.087 (0.011) ** |
| Maternal education (ref. ISCED 0–2) |  |  |  |  |
| ISCED 3–5 | - | 0.242 (0.028) ** | 0.283 (0.035) ** | 0.279 (0.035) ** |
| ISCED 6 | - | 0.401 (0.030) ** | 0.457 (0.036) ** | 0.447 (0.036) ** |
| ISCED 7–8 | - | 0.541 (0.045) ** | 0.605 (0.053) ** | 0.584 (0.054) ** |
| Paternal education (ref. ISCED 0–2) |  |  |  |  |
| ISCED 3–5 | - | 0.187 (0.029) ** | 0.197 (0.036) ** | 0.196 (0.037) |
| ISCED 6 | - | 0.323 (0.032) ** | 0.319 (0.039) ** | 0.313 (0.039) |
| ISCED 7–8 | - | 0.512 (0.040) ** | 0.492 (0.047) ** | 0.483 (0.048) |
| Maternal education x Parental divorce |  |  |  |  |
| ISCED 3–5 x Parental divorce | - | - | -0.120 (0.060) * | -0.119 (0.060) * |
| ISCED 6 x Parental divorce | - | - | -0.176 (0.064) ** | -0.175 (0.064) ** |
| ISCED 7–8 x Parental divorce | - | - | -0.212 (0.105) * | -0.209 (0.105) * |
| Paternal education x Parental divorce |  |  |  |  |
| ISCED 3–5 x Parental divorce | - | - | -0.027 (0.059) | -0.029 (0.059) |
| ISCED 6 x Parental divorce | - | - | 0.023 (0.068) | 0.016 (0.068) |
| ISCED 7–8 x Parental divorce | - | - | 0.096 (0.092) | 0.091 (0.092) |
| Income measures |  |  |  |  |
| Household EDI | - | - | - | -0.006 (0.009) |
| Net income father | - | - | - | 0.006 (0.004) |
| Net income mother | - | - | - | 0.011 (0.006) * |
| Constant | 4.181 (0.015) ** | 3.636 (0.033) ** | 3.593 (0.040) ** | 3.605 (0.040) ** |
| Adjusted $R^2$ | 0.042 | 0.119 | 0.120 | 0.120 |

$b$ = unstandardized regression coefficient, S.E = standard error, ref. = reference group.

Age and income variables are centered on their respective means. All income measures are presented in 100,000 NOK.

EDI = Equivalized disposable income.

* $p < 0.05$

** $p < 0.01$

heterogeneity in the associations between divorce and GPA by mother's or father's net income were found (results not shown).

Of note, it is highly likely that the potential heterogeneity by measures of household income in the links between divorce and academic outcomes is sensitive to how income is operationalized. As parental education was the main focus of interest in the present study, we did not examine this any further in the present paper (e.g., other ways of dividing income into categories).

## Discussion

The purpose of this study was to investigate the possible existence of heterogeneity in the association between divorce and adolescents' academic achievement within the context of an elaborate welfare state such as Norway. As expected, adolescents with divorced parents had on average lower GPA compared to their peers with nondivorced parents. This difference was robust and only moderately reduced (from 0.30 to 0.24 points, about 20%) after adjustments

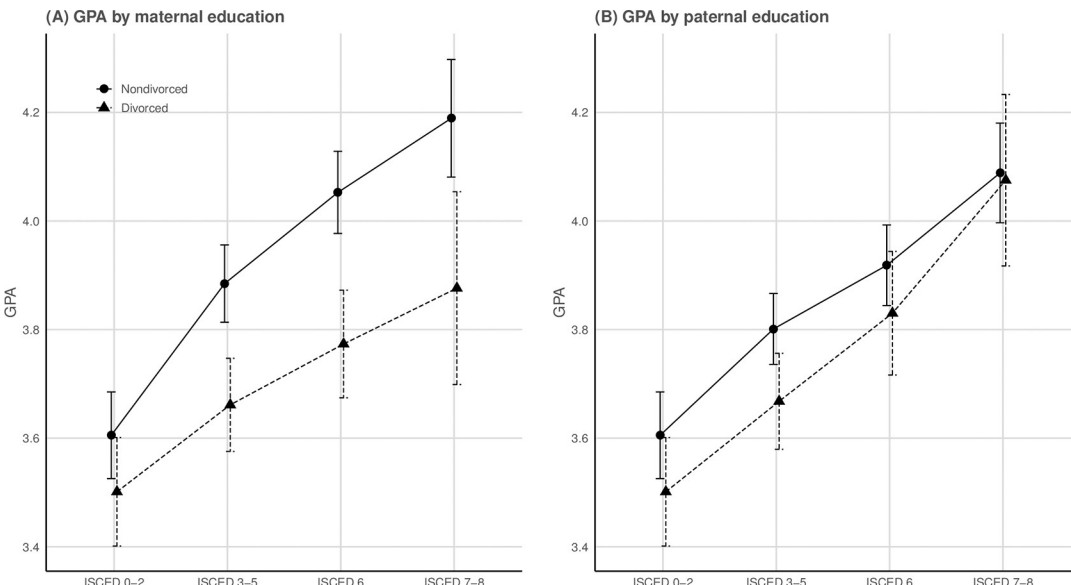

**Fig 2. Predicted values of GPA by maternal and paternal educational qualifications (*n* = 7,739).** The predicted values of GPA by (**A**) maternal educational qualifications, and (**B**) paternal educational qualifications from the fully adjusted regression models, stratified by divorce status (cf. Table 4). Error bars represent 95% confidence intervals of *b*. ISCED 0–2 = up to lower secondary education, ISCED 3–5 = upper secondary education, post-secondary non-tertiary education, short-cycle tertiary education, ISCED 6 = Bachelor's level, ISCED 7–8 = Master's or Doctoral level.

for parental education. This finding fits well with previous studies [2,4,6], and suggest that divorce is associated with poorer school performance also among Norwegian adolescents.

The negative association between divorce and GPA was relatively stronger among adolescents where at least one of the parents had educational qualifications equivalent to upper secondary education or a bachelor's degree, compared to families without any educational qualifications. Further analyses revealed that this heterogeneity was primarily driven by maternal educational levels, whereby having divorced parents was more strongly and negatively associated with the GPA among adolescents with educated mothers (i.e., above ISCED 2) than among adolescents with less educated mothers (i.e., ISCED 0–2), after holding the effects of paternal education and income constant.

Overall, our findings thus lend support to the *floor effects* hypothesis [21], suggesting that the negative associations between divorce and adolescents' academic achievement are relatively stronger among adolescents with educated parents, compared to those with less educated parents. Similar floor effects have been reported in terms of youths' later educational attainment [16,21,22]. Our results also align with the study by Martin [23], which found that the link between divorce and subject grades was relatively stronger among adolescents with educated mothers compared to less educated parents. Other studies have, however, reported either no heterogeneity in GPA according to parental education [21], or a compensatory advantage of having educated parents on current school performance [17,20].

The heterogeneity in the associations between parental divorce and adolescents' GPA observed in the present study stem from two related findings: Firstly, a divorce was hardly related to the GPA among adolescents with two uneducated parents. Secondly, among adolescents with educated or highly educated mothers, a relatively larger negative association between divorce and the adolescents' GPA was observed. We offer the following interpretation of how this pattern may come about within the Norwegian context:

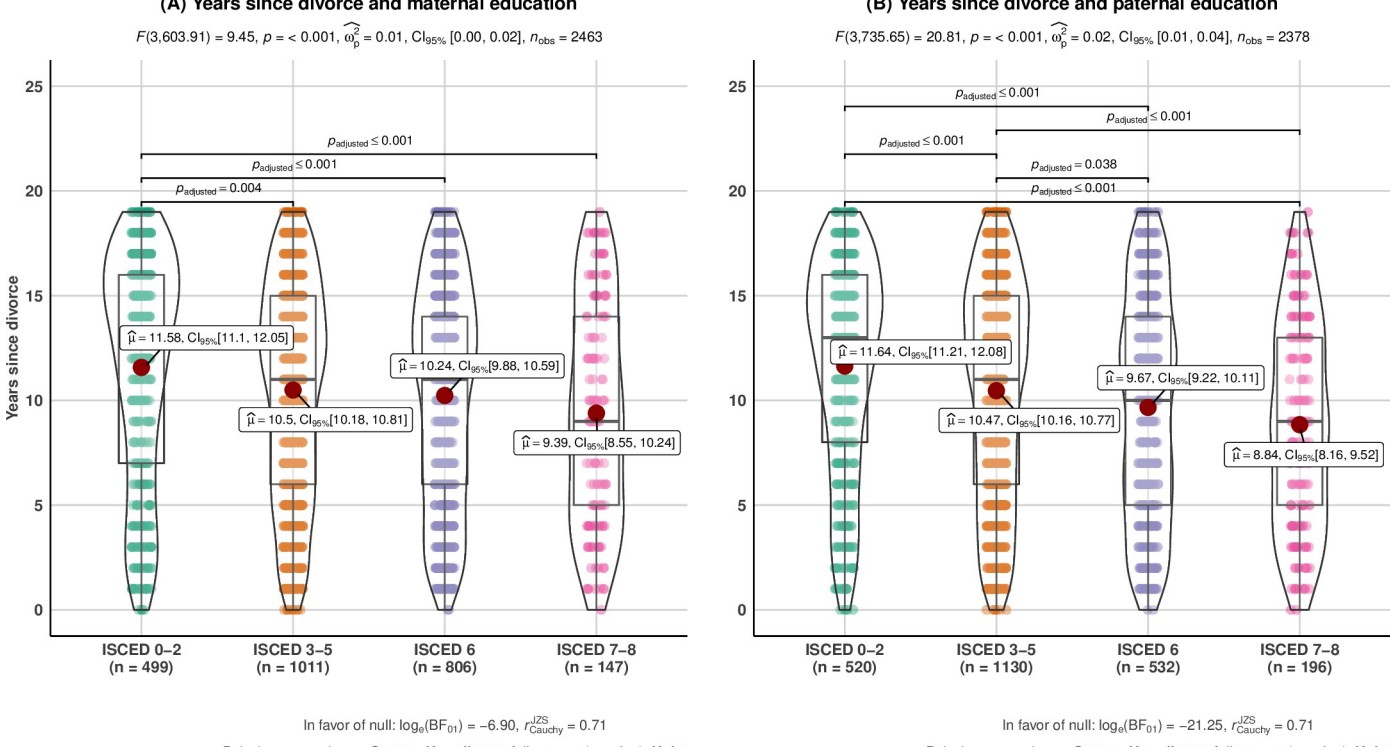

**Fig 3. Years since divorce across maternal ($n = 2,463$) and paternal ($n = 2,378$) educational qualifications.** Pairwise comparisons of years since divorce across maternal (**A**) and paternal (**B**) educational qualifications. The plots comprise a mix of a violin plot (displaying the shape of the variable distribution) and a box plot (where the box is split by the median and bounded by the first and third quartiles of the distribution) along with the jittered raw datapoints. The red dot signifies mean values, also reported in text as $\hat{\mu}$ with accompanying 95% confidence intervals ($CI_{95\%}$). Only significant pairwise comparisons are shown with accompanying p-values.

The Norwegian population is highly educated, and families where both parents have low levels of educational qualifications are relatively rare. Social gradients in health and education are nevertheless well established also in Norway [59,60], and socioeconomically disadvantaged families have more frequent experiences of negative life events and family stresses (e.g., stress related to unemployment, work, and housing) including higher rates of marital dissolution than more affluent families [34,61]. The on average higher levels of family instability experienced by children in uneducated families might suggest that these children have come to expect adverse events in their lives. A divorce might thus be but one of several potential adversities experienced during childhood, rendering the independent effect of divorce less severe [24]. Moreover, the expected school performance among adolescents with parents with low levels of education are, on average, already weak to begin with. Thus, there is less room for their grades deteriorating further as a consequence of divorce. It is also conceivable that the elaborate welfare systems in Norway effectively buffer against further financial strain following divorce among less educated families, perhaps partly because there is less potential for their economic situation to worsen any further. Thus, although less educated families are more likely to experience stress related to poor family finances, a divorce might not exacerbate their financial situation.

We found that the negative association between divorce GPA was relatively larger among adolescents with educated compared to less educated mothers. Martin [23] reported a similar finding in the U.S., which in turn was partly explained by children of educated divorced/single

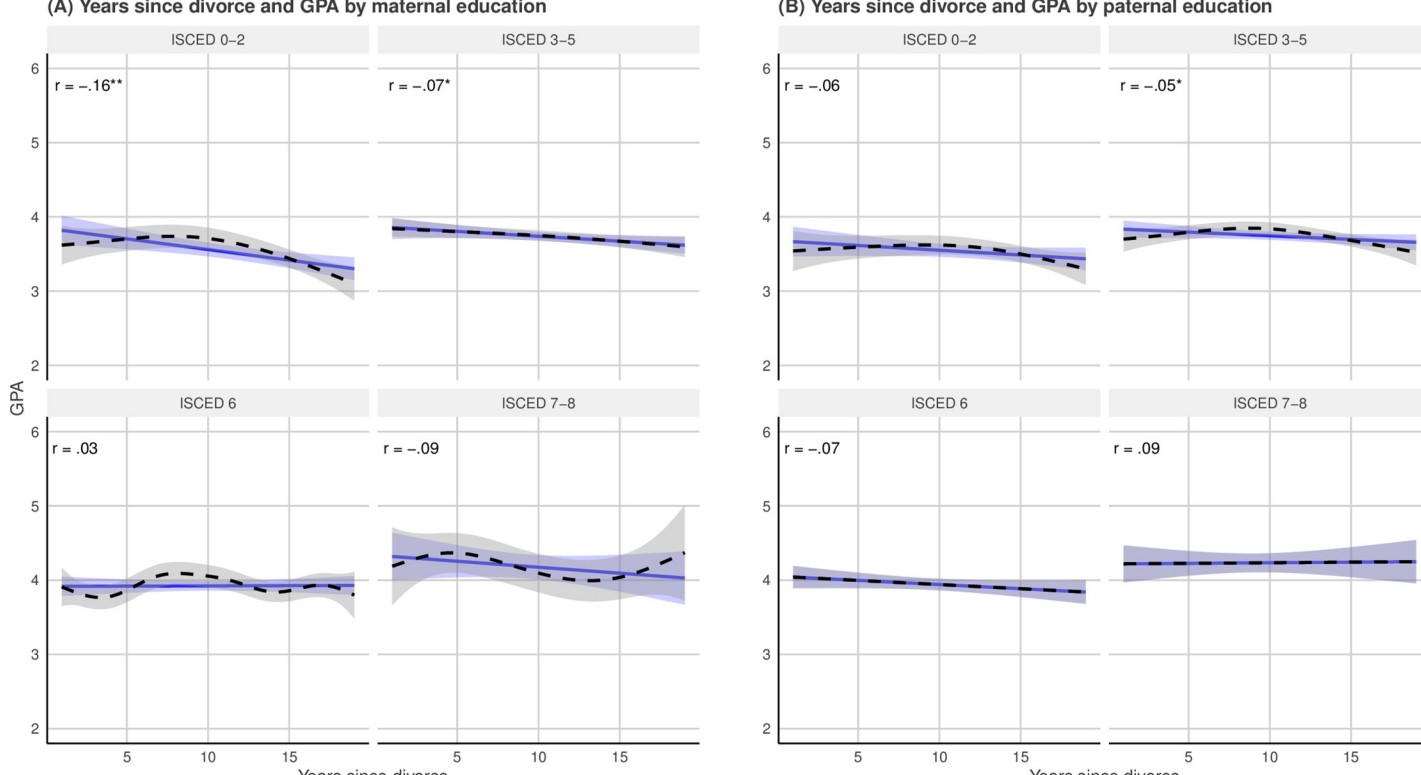

**Fig 4. Associations between years since divorce and GPA by maternal (*n* = 2,463) and paternal (*n* = 2,378) educational qualifications.** This figure shows the associations between years since divorce and the adolescents' GPA by maternal (**A**) and paternal (**B**) educational qualifications. The blue lines represent the linear association, while the smoothed dotted black lines stem from GAMs The shaded area represents the 95% confidence intervals. Pearson's product-moment correlations of the linear associations are displayed in the top left quadrants with asterisks denoting statistically significant associations (* < 0.05, ** <0.01).

mothers not receiving similar levels of positive parenting practices as peers with educated married parents. Such mechanisms were not explored in the current study. Thus, we can only speculate to whether they also apply to the Norwegian context. However, this explanation may fit the notion of a "double burden" experienced by educated, divorced single mothers in Norway, due to the strain of high workload combined with child-rearing responsibilities [37,38]. As it takes time and effort to engage in parenting practices that foster academic skills in children, educated divorced mothers are perhaps less able to continue providing such parenting practices to the same extent after the divorce relative to their equally educated married counterparts. Less educated mothers may, however, not have spent as much time fostering such skills to begin with.

Moreover, as the divorce might come as more of a shock among both parents and children in educated families [24], school-promoting activities might be more offset as the family tries to adjust to post-divorce family life.

Adjusting the analyses by measures of parental income hardly changed the estimated interactions between divorce and parental educational levels on the adolescents' GPA. This contrasts the findings of Bernardi and Boertien [21,22], which found that the link between divorce and adolescents' educational attainment among youth with highly educated parents was driven by the loss of access to father's financial resources following divorce. These studies were, however, conducted on a cohort born in 1970 in Britain, a context with higher levels of single mother poverty and where divorced fathers often failed to pay child support, as noted by the

authors [22]. The Norwegian welfare state, on the other hand, appears to be rather successful in equalizing the cost of divorce among men and women [35,36]. Hence, the departure of an educated father after a divorce may arguably be less economically detrimental to divorced mothers in Norway. Moreover, income is likely more important when considering later educational attainment than current school performance in countries where admission to higher education is costly. It is important to stress that our findings do not suggest that income is irrelevant to children's post-divorce adjustment, and indeed, single mothers are also in Norway among the least well of in society. Instead, the results of this study suggest that other mechanisms might be more important in explaining the observed heterogeneity in the associations between divorce and GPA by parental educational qualifications.

Our findings were robust to adjustments of age, gender, and current income measures. Adolescents with highly educated parents experienced, on average, that their parents had divorced somewhat more recently. Time since divorce was overall weakly and negatively associated with the adolescents' GPA, with some small observable variations across parental educational qualifications. These variations could, however, not explain the heterogeneity found in the present study. Overall, this is generally in keeping with studies that find that youth post-divorce adjustment is quite stable or gradually slightly worsen as time passes since the divorce [62,63].

Some previous studies have adjusted their analyses by different pre-divorce characteristics such as the child's behavioral problems, cognitive abilities, and material resources [21], the child's psychological well-being, academic ability, mothers' distress, and their family's pre-divorce economic resources [20], or utilized family fixed effects models [17]. Unfortunately, besides parental education, which is usually established before a divorce, other pre-divorce measures were not available in the current study. It is plausible that the differential inclusion of pre-divorce characteristics may account for parts of the diverging results. Other differences between the studies, such as the operationalization of dependent variables, differences in age groups, and cross-national differences, are also likely of importance. Compared to previous studies examining cohorts from the 1970s - 1980s [20,21,23], the current study is nonetheless unique by being based on a relatively recent cohort (born 1993–1995, and assessed in 2012). Due to the elaborate social welfare systems in Norway, where the school system is highly subsidized, higher education is common, and gender equity is high [44,64], direct comparisons with studies utilizing older samples from other countries should be made with care.

The following limitations of the current study should further be acknowledged. Firstly, due to the cross-sectional structure of our study, we have had no means of investigating potential changes from pre- to post-divorce family life. Proposed explanations such as changes in parenting among highly educated parents following divorce have, therefore, not been examined. Hence, this study is largely descriptive. Similarly, we did not have data to control for further differences between divorced and nondivorced families, which may induce selection effects [1]. For example, historical information that could both increase parents' inclination to divorce, and possibly affect the adolescents' GPA differently according to parental educational levels (e.g., mental health problems or levels of parental conflict) could be of importance and might explain the larger educational penalty observed in families with highly educated parents [22,65]. Of note, studies that have statistically adjusted their analysis by pre-divorce characteristics often find that it weakens the associations between divorce and educational outcomes [43]. The present study might thus overestimate the link between divorce and GPA. Nonetheless, divorce is generally understood as a process that gradually unfolds, rather than being a discrete point in time [1]. Adjustments of pre-divorce characteristics are thus not without problems either, as it may remove some of the effects that are intrinsically linked to the divorce

process. Adjustments of pre-divorce characteristics should, therefore, be made with care [1,66].

The present study focused on the distinction between divorced and nondivorced families, whereas an investigation of other family structures or arrangements (e.g., single father families, stepfamilies, joint physical custody) were outside the scope of this study. Adolescents' mental health and school engagement are found to vary across family structures [67–69]. It would be interesting for future studies to conduct a more detailed investigation of whether heterogeneous outcomes of divorce by parental education also depend on the post-divorce family structure. The findings of the present study might also depend on the adolescents' gender, an issue that warrants further investigations.

Lastly, the participation rate in the youth@hordaland study was 53%, and the sample in the current study was further reduced to 47% of the total invited population. Previous investigations have found that the GPA in this sample were quite similar to and not statistically significantly different from both the regional and national averages [46]. Nevertheless, non-response is known to be related to lower socioeconomic status, and an earlier study on the former waves of the Bergen Child Study (which the youth@hordaland is nested within) found more psychological problems among those not participating [70]. This could limit the generalizability of the findings.

Despite these limitations, some strengths of the current study deserve mentioning. We utilized high-quality register-based information about the adolescents' GPA, parental educational levels, and household income, which was merged with a large population-based study. Whereas previous studies have calculated the GPA by a subset of either subject grades, test scores, or exams, sometimes self-reported, the measure of GPA utilized in the current study is calculated from all graded subjects during a whole school year. As the measure of GPA used in the present study forms the primary basis for admittance into higher education in Norway, it may be considered highly reliable.

## Conclusion

To conclude, the present study found that the association between parental divorce and adolescents' GPA is robust also within a Norwegian context. However, divorce was hardly associated with GPA among adolescents whose parents have low educational qualifications. In contrast, among adolescents from families with educated or highly educated mothers, parental divorce was associated with a lower GPA. These findings were robust to adjustments of measures of household income. Future studies are needed to investigate potential mechanisms (such as reduced parental monitoring or school-involvement), which might drive this finding.

The generalizability of these findings might be limited to a Norwegian context, as differences in both school systems and policies across nations may play an essential part in how parental divorce and parental education might affect adolescents' academic performance. Due to diverging results among existing studies examining this phenomenon, there is a need for future studies that can shed further light on the complex interactions between divorce, parental education, and outcomes among youths.

## Supporting information

**S1 Fig. Regression coefficient plot of estimates of GPA by parental divorce and equivalized disposable income (n = 7,739).**
(PDF)

**S1 Table. Comparison of the entire youth@hordaland with adolescents with register-based information.**
(PDF)

## Author Contributions

**Conceptualization:** Sondre Aasen Nilsen, Kyrre Breivik, Bente Wold, Børge Sivertsen, Mari Hysing, Tormod Bøe.

**Data curation:** Sondre Aasen Nilsen, Kristin Gärtner Askeland, Mari Hysing, Tormod Bøe.

**Formal analysis:** Sondre Aasen Nilsen, Kyrre Breivik, Tormod Bøe.

**Funding acquisition:** Sondre Aasen Nilsen, Tormod Bøe.

**Methodology:** Sondre Aasen Nilsen, Kyrre Breivik, Kristin Gärtner Askeland, Mari Hysing, Tormod Bøe.

**Software:** Sondre Aasen Nilsen.

**Supervision:** Kyrre Breivik, Bente Wold, Tormod Bøe.

**Visualization:** Sondre Aasen Nilsen.

**Writing – original draft:** Sondre Aasen Nilsen.

**Writing – review & editing:** Sondre Aasen Nilsen, Kyrre Breivik, Bente Wold, Kristin Gärtner Askeland, Børge Sivertsen, Mari Hysing, Tormod Bøe.

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
