## [Decision Letter · Decision Letter 0]

21 Oct 2019

PONE-D-19-20535

Divorce and adolescent academic achievement: The moderating role of parental education

PLOS ONE

Dear PsyD Nilsen,

Thank you for submitting your manuscript to PLOS ONE. After careful consideration, we feel that it has merit but does not fully meet PLOS ONE’s publication criteria as it currently stands. Therefore, we invite you to submit a revised version of the manuscript that addresses the points raised during the review process.

Thank you very much for your patience. I really enjoyed reading your paper, as it brings many exciting results, and I am sure that it will enrich the current knowledge on the topic investigated in your study. 

Also, both reviewers found your paper interesting and acknowledge its potential. The reviewers are not entirely convinced that your analysis fully supports all your claims (particularly Reviewer 1). Reviewer 1 further suggests an alternative framework for capturing the effects of parental divorce on test scores. Could you please consider their suggestions? 

I further believe that addressing the rest of the reviewers' comments should not be too challenging. 

We would appreciate receiving your revised manuscript by Dec 05 2019 11:59PM. To enhance the reproducibility of your results, we recommend that if applicable you deposit your laboratory protocols in protocols.io, where a protocol can be assigned its own identifier (DOI) such that it can be cited independently in the future. For instructions see: http://journals.plos.org/plosone/s/submission-guidelines#loc-laboratory-protocols

We look forward to receiving your revised manuscript.

Kind regards,

Tomáš Želinský, Ph.D.

Academic Editor

PLOS ONE

**Journal Requirements:**

**Comments to the Author**

1. Is the manuscript technically sound, and do the data support the conclusions?

Reviewer #1: Partly

Reviewer #2: Partly

2. Has the statistical analysis been performed appropriately and rigorously? 

Reviewer #1: No

Reviewer #2: Yes

3. Have the authors made all data underlying the findings in their manuscript fully available?

Reviewer #1: Yes

Reviewer #2: No

4. Is the manuscript presented in an intelligible fashion and written in standard English?

Reviewer #1: Yes

Reviewer #2: Yes

5. Review Comments to the Author

Reviewer #1: Dear authors,

Thank you for the opportunity to read and comment on your work. I will proceed by addressing the recommended points mentioned in PLOS reviewer guidelines.

What are the main claims of the paper and how significant are they for the discipline?

The paper claims to offer evidence of heterogeneous effects of parental divorce on adolescent GPA. This is a relevant question that has clearly connections to recently published peer reviewed work.

Are the claims properly placed in the context of the previous literature? Have the authors treated the literature fairly?

The claims are properly placed, and I feel that the authors treated the literature fairly.

Do the data and analyses fully support the claims? If not, what other evidence is required?

-I do not believe that the current analysis offers the evidence needed to support the stated claims. The present analysis interacts a binary indicator of parental divorce experience with various measures of parental education in OLS regressions predicting mean GPA of children. The results successfully show that the difference in GPA between kids from surviving marriages and kids of divorce tends to be larger when mothers are more educated. While I agree with this result, I do not believe that this is the result that you need to support your claim.

When considering the effects of parental divorce on test scores, I conceptualize a framework where a student has produced some GPA history up until divorce occurs at time t=0. The disruption occurs at time zero, and in subsequent periods the GPA realization may be heavily affected by the disruption, but decreasingly affected over time until the effect dies or becomes undetectable. In this framework, parental divorce should have a measurable effect on GPA in a limited time period after the event. You would then need to show that the negative disruption relative to the t≤0 levels is larger for kids with more educated mothers/parents to support your point. This type of analysis would require data on the timing of divorce, as well as some type of fixed effects model that produces estimates using within person variability in GPA between pre- and post-divorce periods.

This approach would be feasible if your two divorce indicator questions were asked consistently from year to year of your longitudinal survey. If this is the case, you can easily write code that identifies the year in which kids experience divorce allowing the formation of a clear partition between pre- and post-divorce GPA observations.

-I also have a question about the timing of study events. Were kids asked about divorce experience before, or after the GPA observations. If kids were asked about divorce experience before GPA realizations, is it possible to confirm that there are not kids in the non-divorce group who divorce during GPA observations? If GPA is observed before posing the divorce questions, how do you know which observations occurred before the divorce and which occurred after?

-Can you say anything about differences in the timing of divorce between more and less educated couples? I would expect more educated couples to divorce later, on average. If this is the case, the larger effects for more educated parents might be catching divorce experiences that are closer to the GPA measurement dates. This would suggest an alternate story where effects may not be stronger for more educated families. Instead, we may be measuring effects for more educated families when effects are largest, and measuring effects for less educated families when effects are already subsiding.

-This project seems somewhat limited by the OLS modeling choice, with the constraints of your data. The OLS approach likely over-estimates divorce effects by adding co-linear effects of the unobserved factors that lead to divorce. You would need a very rich set of controls to convincingly isolate the divorce effect in this framework. Unfortunately, your data do not appear to offer richer controls that would add more precision to the divorce estimates.

PLOS ONE encourages authors to publish detailed protocols and algorithms as supporting information online. Do any particular methods used in the manuscript warrant such treatment? If a protocol is already provided, for example for a randomized controlled trial, are there any important deviations from it? If so, have the authors explained adequately why the deviations occurred?

No.

If the paper is considered unsuitable for publication in its present form, does the study itself show sufficient potential that the authors should be encouraged to resubmit a revised version?

I do not know. I believe that the authors would need to write a different paper with a different statistical analysis to answer their stated question. At the same time, I feel that the work that they have done thus far is interesting, useful, and publishable if the results were directed toward a different component of their original question.

I see more than one way of doing this, but my immediate suggestion would be to follow through more fully on your interest in understanding how the Norwegian context matters for these effect estimates. Given that your OLS model is limited in its ability to isolate divorce effects, compare your estimates to other work that faces the same challenge and make an empirically motivated theoretical contribution about the relevance of national and cultural context in determining divorce effects after comparing and contrasting your estimates with results from comparable studies in different contexts.

Are original data deposited in appropriate repositories and accession/version numbers provided for genes, proteins, mutants, diseases, etc.?

I believe so.

Are details of the methodology sufficient to allow the experiments to be reproduced?

I believe so.

Is the manuscript well organized and written clearly enough to be accessible to non-specialists?

Yes.

Reviewer #2: Thank you for giving me an opportunity to read your paper. I think it is a strong study. In particular, I find the discusssion section strong. Nevertheless I have some comments, which I hope you find useful.

1. As the authors note themselves on page 27 their study is descriptive. Therefore any causal language (effects, impacts, penalty, etc.) should be avoided. The same goes for the "moderating" role of parental education in the title. Parental education could also pick up the effect of some other, unobserved variable. It is quite easy to think that socioeconomic differences in the effects of divorce/ separation on children's school grades may be very different from socioeconomic differences in the associations between divorce/ separation and children's school grades. (I also do not thin that proposensity score approaches should be called causal as the authors do on page 3; the crucial step in causal analysis is to control for selection on unobserved variables.)

2. The paper uses survey data on Norway, a country in which register data is available and has been previously used (e.g., Sigle-Rushton et al., 2014). Why not use data on the whole population?

3. It would be nice to see some robustness checks with other indicators of social origin, e.g. parental occupation, income, and wealth. Also, what happens if the reference category of parental education is changed? Currently it seems as differences are mainly due to ISCED 0-2 vs. ISCED 3-7 (Model 2 in Table 2). As the authors note on page 24, the ISCED 0-2 is a disdvantaged group. Therefore, ISCED 3-4 may be a better choice for a reference category.

4. Condition on household income introduces overcontrol bias (model 3). The authors may discuss this issue. (Although I understand as well that the results are rarely affected by this control/ mediator.) There is also a problem of overcontrol bias when conditioning on paternal and maternal education simultaneously (Model 2 in Table 4). Can the authors enter only maternal education and then only paternal education in one set of models and do they get the same results?

5. Did the authors look at gender differences in the associations?

Smaller points:

- The authors study the dissoultion of married and non-married couples. Maybe therefore better to use the term separation throughout the whole study? The authors note on page 13 that the research practice is to use divorce but it does not strike me too be a good practice.

- Given that grades are obtained in different school years, may it not make sense to rank or standardize the variable within school years?

- The authors should add sample sizes to all tables and figures.

6. PLOS authors have the option to publish the peer review history of their article (what does this mean?). If published, this will include your full peer review and any attached files.

Reviewer #1: Yes: Ravaris Moore

Reviewer #2: No

---

## [Author Response · Author response to Decision Letter 0]

11 Nov 2019

Author Response to Reviewers’ Comments

Journal: PLOS ONE

Manuscript ID: PONE-D-19-20535

Dear Tomáš Želinský, Ph.D.

Thank you for allowing us to revise and resubmit our paper. Below are the comments from the reviewers (numbered) and our responses. We have uploaded one version of the manuscript where these changes are identified using track changes (Revised Manuscript with Track Changes”) and one version that is clean (Manuscript).

We thank you for your consideration of our revised manuscript and responses, and we look forward to hearing from you.

Sincerely,

Sondre Aa. Nilsen

(on behalf of all authors)

 

Response to Editorial comments:

1. Thank you very much for your patience. I really enjoyed reading your paper, as it brings many exciting results, and I am sure that it will enrich the current knowledge on the topic investigated in your study. Also, both reviewers found your paper interesting and acknowledge its potential. The reviewers are not entirely convinced that your analysis fully supports all your claims (particularly Reviewer 1). Reviewer 1 further suggests an alternative framework for capturing the effects of parental divorce on test scores. Could you please consider their suggestions? I further believe that addressing the rest of the reviewers' comments should not be too challenging.

Response: We thank the editor for these positive comments and for sharing our enthusiasm with regards to this research topic. We have made several changes to the manuscript in response to the suggestions made by the reviewers. Their comments and our point-by-point responses can be found below. 

Response to Comments from Reviewer 1

We thank Reviewer 1 for the evaluation of the manuscript, and for providing suggestions for improvements. 

Reviewer comments:

1. What are the main claims of the paper and how significant are they for the discipline?

The paper claims to offer evidence of heterogeneous effects of parental divorce on adolescent GPA. This is a relevant question that has clearly connections to recently published peer reviewed work.

Response: Thank you for this comment.

2. Are the claims properly placed in the context of the previous literature? Have the authors treated the literature fairly?

The claims are properly placed, and I feel that the authors treated the literature fairly.

Response: Thank you for stating that our claims are properly placed, and that we have treated the literature fairly. 

3. Do the data and analyses fully support the claims? If not, what other evidence is required?

a. -I do not believe that the current analysis offers the evidence needed to support the stated claims. The present analysis interacts a binary indicator of parental divorce experience with various measures of parental education in OLS regressions predicting mean GPA of children. The results successfully show that the difference in GPA between kids from surviving marriages and kids of divorce tends to be larger when mothers are more educated. While I agree with this result, I do not believe that this is the result that you need to support your claim.

Response: We fully agree with the reviewer that due to limitations with our data (i.e., the lack of repeated/longitudinal measures of GPA in the years before and after the divorce), we have had no means to try to approximate the causal relationship between divorce, parental educational qualifications, and the adolescents’ GPA. We do see that our manuscript contains some usage of causal language, which clearly should be avoided (as also pointed out by Reviewer 2, point 1). We have therefore made changes to the title and throughout the manuscript in order to stress that this is a descriptive association study, and that any offered explanations for our findings are based on what we consider theoretically or empirically relevant information from other studies. We have made it clearer throughout the manuscript that the main aim of the present study was to examine potential heterogeneity in the associations between parental divorce and adolescents’ GPA by parental educational qualifications and also note the descriptive nature of our study in the limitations section (page 28, lines 594 - 595). 

The title now reads: 

Divorce and adolescent academic achievement: Heterogeneity in the associations by parental education

b. When considering the effects of parental divorce on test scores, I conceptualize a framework where a student has produced some GPA history up until divorce occurs at time t=0. The disruption occurs at time zero, and in subsequent periods the GPA realization may be heavily affected by the disruption, but decreasingly affected over time until the effect dies or becomes undetectable. In this framework, parental divorce should have a measurable effect on GPA in a limited time period after the event. You would then need to show that the negative disruption relative to the t≤0 levels is larger for kids with more educated mothers/parents to support your point. This type of analysis would require data on the timing of divorce, as well as some type of fixed effects model that produces estimates using within person variability in GPA between pre- and post-divorce periods.

This approach would be feasible if your two divorce indicator questions were asked consistently from year to year of your longitudinal survey. If this is the case, you can easily write code that identifies the year in which kids experience divorce allowing the formation of a clear partition between pre- and post-divorce GPA observations.

Response: Thank you for this insightful comment. We agree that this would be a feasible way to analyze longitudinal data with repeated measures of GPA and information about the timing of divorce, in order to come closer to establish a causal relationship. This would be a very interesting study to conduct. To the best of our knowledge, very few studies investigating heterogeneity in the link between divorce and adolescents’ GPA by parental education have had such rich longitudinal data available (although studies have to various degrees been able to control for some pre-divorce characteristics). Unfortunately, our study is also limited in this sense. The youth@hordaland study, is a cross-sectional study that was merged with register-based information on parental education and subject grades obtained during the school year 2011-2012. As such, we had no means to track adolescents’ GPA from pre- to post-divorce times (as noted on page 27 – 28, lines 591-594). 

To make this more explicit, we have updated both the Abstract and the Procedure sections of the manuscript, to highlight that this study draws on cross-sectional data. 

c. -I also have a question about the timing of study events. Were kids asked about divorce experience before, or after the GPA observations. If kids were asked about divorce experience before GPA realizations, is it possible to confirm that there are not kids in the non-divorce group who divorce during GPA observations? If GPA is observed before posing the divorce questions, how do you know which observations occurred before the divorce and which occurred after?

Response: The youth@hordaland study was conducted during spring of 2012, while the GPA was calculated by all grades obtained during the first term of the school year 2011/2012. Although possible, we find it unlikely that many experienced that their parents divorced during this short period, at least not in sufficient number to lead to any substantial change in the divorce – non-divorced categorization. 

As shown in Table 1, the mean year since experiencing parental divorce was 10.58 years. Only16 respondents stated that their parents had divorced during the current school year. The rest reported that at least one year had gone since their parents divorced. As further elaborated in our response to reviewer comment 3d, a very weak relationship between the timing of divorce and the adolescents’ GPA was found, and accounting for the year of parental divorce did not alters the link between parental educational qualifications and the adolescents’ GPA among adolescents with divorced parents.

d. -Can you say anything about differences in the timing of divorce between more and less educated couples? I would expect more educated couples to divorce later, on average. If this is the case, the larger effects for more educated parents might be catching divorce experiences that are closer to the GPA measurement dates. This would suggest an alternate story where effects may not be stronger for more educated families. Instead, we may be measuring effects for more educated families when effects are largest, and measuring effects for less educated families when effects are already subsiding.

Response: Thank you for raising this interesting point. Indeed, adolescents with educated parents, did, on average, report that their parents had somewhat more recently divorced than less educated parents. However, most adolescents generally report quite some time since parental divorce, which is not too surprising given that we examine rather old adolescents aged 16 – 19. If considering the highest completed education in the family variable (i.e., either by the mother or father), mean years since divorce was 9.1 years for adolescents from the most educated families (i.e.,ISCED 7-8), while mean years since divorce was 12.18 for adolescents with the least educated parents (i.e., ISCED 0 -2). For those with highest education equivalent to ISCED 3-5 and ISCED 6, the average years since divorce was 10.58 and 10.23, respectively (Significant differences found between all educational levels except ISCED 3-5 and ISCED 6 in years since divorce).

In our data, we find a very weak and slightly curvilinear relationship between time since divorce and the adolescents GPA; the GPA rises slightly from 0 to about 8 years since divorce, then gradually declines again from 8 to 19 years since divorce. Overall, this trend nonetheless suggests that the lower GPA among adolescents with divorced compared to nondivorced parents appear rather stable irrespectively of time since divorce. This finding is in general keeping with previous literature suggesting that children’s adjustment following divorce is rather stable but somewhat worse than their peers with non-divorced parents [see e.g., 1].

1. Härkönen J, Bernardi F, Boertien D. Family dynamics and child outcomes: An overview of research and open questions. Eur J Popul. 2017;33

Adding time since divorce in subgroup analyses on adolescents with divorced parents did not in any substantial way alter the link between parental educational qualifications and the adolescents’ GPA. Thus, time since divorce does not appear to account for the heterogeneity in the associations between divorce and GPA by parental education found in the present study. We thank the reviewer for raising this interesting point, and we have added an extra paragraph to the statistics section: 

The timing of divorce could potentially covary with parental educational levels and the adolescents’ GPA (e.g., if highly educated parents divorced later on, the estimates of divorce by parental education on the adolescents’ GPA could pick up on the proximity to the event of dissolution). Additional checks were therefore made to investigate whether the estimates were influenced by the timing of divorce (as measured by years since the divorce). 

And added some extra results to the result section: 

Adolescents with highly educated parents experienced, on average, that their parents divorced somewhat later (mean years since divorce across highest parental education in the family; ISCED 7-8 = 9.1 years, ISCED 6 = 10.2 years, ISCED 3-5 = 10.6 years, ISCED 0-2 = 12.2 years). Subgroup analyses showed a very weak and slightly curvilinear association between years since the divorce and the adolescents GPA; the GPA rose slightly from 0 to about 8 years since divorce, then gradually declined from 8 to 19 years since divorce. Overall, this trend nonetheless suggested considerable stability in the negative association between divorce and adolescents’ GPA. The association between parental education and GPA among adolescents with divorced parents hardly changed when adjusted by years since divorce (results not shown). 

e. -This project seems somewhat limited by the OLS modeling choice, with the constraints of your data. The OLS approach likely over-estimates divorce effects by adding co-linear effects of the unobserved factors that lead to divorce. You would need a very rich set of controls to convincingly isolate the divorce effect in this framework. Unfortunately, your data do not appear to offer richer controls that would add more precision to the divorce estimates.

Response: We agree that our data limits our ability to draw causal conclusions, and that one would need a very rich set of controls to isolate the divorce effects. The fact that the OLS approach might over-estimate the link between divorce and the adolescents’ GPA have been acknowledged as a limitation in the manuscript (page 28, line 603). 

f. I believe that the authors would need to write a different paper with a different statistical analysis to answer their stated question. At the same time, I feel that the work that they have done thus far is interesting, useful, and publishable if the results were directed toward a different component of their original question.

g. I see more than one way of doing this, but my immediate suggestion would be to follow through more fully on your interest in understanding how the Norwegian context matters for these effect estimates. Given that your OLS model is limited in its ability to isolate divorce effects, compare your estimates to other work that faces the same challenge and make an empirically motivated theoretical contribution about the relevance of national and cultural context in determining divorce effects after comparing and contrasting your estimates with results from comparable studies in different contexts.

Response: Thank you for stating that our work is interesting and useful, and for providing further suggestions on how to structure this manuscript. We have updated the aims of the study to highlight that this study sought to investigate heterogeneity in the associations between divorce and GPA by parental educational levels, and made several changes to the manuscript including the title and the abstract, to stress that this is a descriptive, associative study. Finally, we have also done our best to contrast our findings against previous studies in terms on different national and cultural context,

Reviewer #2: Thank you for giving me an opportunity to read your paper. I think it is a strong study. In particular, I find the discussion section strong. Nevertheless I have some comments, which I hope you find useful.

Response: We thank Reviewer 2 for the thorough evaluation of our manuscript, the positive feedback stating that this is a strong study, and for providing several constructive means to improve the manuscript.

1. As the authors note themselves on page 27 their study is descriptive. Therefore any causal language (effects, impacts, penalty, etc.) should be avoided. The same goes for the "moderating" role of parental education in the title. Parental education could also pick up the effect of some other, unobserved variable. It is quite easy to think that socioeconomic differences in the effects of divorce/ separation on children's school grades may be very different from socioeconomic differences in the associations between divorce/ separation and children's school grades. (I also do not think that propensity score approaches should be called causal as the authors do on page 3; the crucial step in causal analysis is to control for selection on unobserved variables.)

Response: Thank you for highlighting this, we fully agree that all causal language should be avoided. We have now made changes throughout the manuscript to address this. We have removed the “propensity score” part in the introduction so to not give the impression that propensity score matching sufficiently addresses causality. We have also updated the title of the manuscript which now reads: 

Divorce and adolescent academic achievement: Heterogeneity in the associations by parental education

2. The paper uses survey data on Norway, a country in which register data is available and has been previously used (e.g., Sigle-Rushton et al., 2014). Why not use data on the whole population?

Response: This study is part of a larger project financed to investigating outcomes of divorce among adolescents utilizing data from the youth@hordaland study. We managed to link the youth@hordaland to official registries, and saw the possibility to capitalize on this linkage in order to investigate heterogenous associations between divorce and adolescents’ GPA by parental educational qualifications. We agree that it would be very interesting to get hold of data of the entire population. In this study, we have, however, been limited to the data at hand from the youth@hordaland study. 

3. It would be nice to see some robustness checks with other indicators of social origin, e.g. parental occupation, income, and wealth. Also, what happens if the reference category of parental education is changed? Currently it seems as differences are mainly due to ISCED 0-2 vs. ISCED 3-7 (Model 2 in Table 2). As the authors note on page 24, the ISCED 0-2 is a disdvantaged group. Therefore, ISCED 3-4 may be a better choice for a reference category.

Response: Unfortunately, we did not have information about parental occupation or wealth that would have enabled us to investigate heterogeneity in the associations between divorce and adolescents’ GPA on these indicators. The existing literature has primarily focused on parental education as a measure of social origin, and our aim with this study was to continue this research tradition within a Norwegian context. We agree that it would be interesting to examine this further with other indicators of social origin in future studies. 

Alternating the reference category of the parental educational variables highlight that the significant differences lies between ISCED 0-2 and the other educational categories. Thank you for noticing that we have not explicitly mentioned this in the manuscript. We have added information about this in the result section:

Alternating the reference categories of the parental education variables in the regression analyses did not reveal any further statistically significant differences in the links between divorce and GPA by parental educational qualifications (i.e., the main differences were between the ISCED 0-2 levels and the other ISCED levels). 

4. Condition on household income introduces overcontrol bias (model 3). The authors may discuss this issue. (Although I understand as well that the results are rarely affected by this control/ mediator.) There is also a problem of overcontrol bias when conditioning on paternal and maternal education simultaneously (Model 2 in Table 4). Can the authors enter only maternal education and then only paternal education in one set of models and do they get the same results?

Response: We thank the reviewer for raising this interesting point. We agree that conditioning on household income could produce overcontrol bias when entered simultaneously with parental education in the same model. Controlling for maternal and paternal education might be problematic for similar reasons, as these are usually dependent (e.g., through “assortative mating”).

We have checked whether we get the same results by entering maternal education and paternal education separately. The results from these analyses are very similar to those reported in the paper. The only notable difference is that maternal education at ISCED 7-8 is not statistical significantly different compared to the ISCED 0-2 level (b = -0.163, p = 0.096), although the estimate is only slightly weaker than when paternal educational qualifications is also added to the model (by 0.04 GPA points). 

We have made the following changes in the statistics section in order to address the issue of overcontrol bias: 

Conditioning on measures of income and paternal and maternal education simultaneously may introduce overcontrol bias [51]. In the two first set of regression analyses, we try to avoid this problem by creating single measures combining information on parental education from both parents, and by entering income variables in the last set of models (as we were not interested in the main effects of income per se). In the last set of regressions (cf. Table 4), we have made robustness checks by entering maternal and paternal education in separate models. 

We have also added some information in the result section: 

Entering maternal and paternal educational levels in separate models yielded approximately identical estimates. The only exception was that the difference in the relationship between divorce and GPA was slightly smaller and not statistically significant different (at p < 0.05) between the highest maternal educational levels (ISCED 7-8) compared to the lowest maternal educational levels (ISCED 0-2) in the interaction analyses (b = -0.163, p = 0.096).

5. Did the authors look at gender differences in the associations?

Response: We agree that it would be of interest to examine potential gender differences in the associations between divorce/separation on GPA by parental education. However, due to limited statistical power, we did not investigate this further in the current study. We see that gender differences remains largely unexplored within this research field, and it would be interesting for future studies to look into this, perhaps using register data with larger samples. 

We have added a sentence in the limitations section of the paper to highlight that we have not investigated gender differences: 

The findings of the present study might also depend on the adolescents’ gender, an issue we did not investigate due to power restrictions.

Smaller points:

- The authors study the dissolution of married and non-married couples. Maybe therefore better to use the term separation throughout the whole study? The authors note on page 13 that the research practice is to use divorce but it does not strike me too be a good practice.

Response: We see that recent studies perhaps tend to utilize the term parental separation rather than divorce. One could argue that the term “parental separation” is not completely satisfying either, as it may be used to denote the status preceding a legal divorce (i.e., in several countries including Norway, one has to be separated a given time period before filing for a legal divorce). 

This paper is part of a larger project whereby the term divorce is consistently used to denote couples who split up from marriage or cohabitation, and we therefore hope to keep this term also in this paper. We have been explicit in this paper with regards to how we operationalize divorce, and we hope that the reviewer might reconsider. If the reviewer feels very strongly about this point, we will nonetheless change the term to separation. 

- Given that grades are obtained in different school years, may it not make sense to rank or standardize the variable within school years?

Response: We have added the adolescents’ age as a covariate in all regression models, a variable that indirectly captured the school year that the adolescents were in at the time of the study (at least for the vast majority of the adolescents). It would make more sense to rank/standardize the variable within school years. However, besides the adolescents age and the year (i.e., 2011/2012) of which the grades were given, we do not have information regarding which specific school year the adolescents were in at the time of the study. We are therefore unable to further rank the variable within school years. 

- The authors should add sample sizes to all tables and figures.

Response: Thank you for noticing, sample sizes have now been added to all tables and figures.

---

## [Decision Letter · Decision Letter 1]

15 Jan 2020

PONE-D-19-20535R1

Divorce and adolescent academic achievement: Heterogeneity in the associations by parental education

PLOS ONE

Dear PsyD Nilsen,

Thank you for submitting your manuscript to PLOS ONE. After careful consideration, we feel that it has merit but does not fully meet PLOS ONE’s publication criteria as it currently stands. Therefore, we invite you to submit a revised version of the manuscript that addresses the points raised during the review process.

Once again, I would like to thank you for your patience. Both referees were happy with your adjustments, and they only suggest minor revisions, which I believe, reflects the quality of the improvements you made. 

Both referees made a point related to the robustness checks, although from a different perspective. While Reviewer 1 would like to see whether adding "a control measure for time since separation/divorce" could affect your main findings, Reviewer 2 would like to see the results for an additional robustness check using household income as an alternative indicator of a family's socioeconomic resources. 

I believe that these suggestions will not affect the main results of your paper, and I will be happy to recommend your paper for acceptance. 

We would appreciate receiving your revised manuscript by Feb 29 2020 11:59PM. To enhance the reproducibility of your results, we recommend that if applicable you deposit your laboratory protocols in protocols.io, where a protocol can be assigned its own identifier (DOI) such that it can be cited independently in the future. For instructions see: http://journals.plos.org/plosone/s/submission-guidelines#loc-laboratory-protocols

We look forward to receiving your revised manuscript.

Kind regards,

Tomáš Želinský, Ph.D.

Academic Editor

PLOS ONE

Reviewers' comments:

Reviewer's Responses to Questions

**Comments to the Author**

1. If the authors have adequately addressed your comments raised in a previous round of review and you feel that this manuscript is now acceptable for publication, you may indicate that here to bypass the “Comments to the Author” section, enter your conflict of interest statement in the “Confidential to Editor” section, and submit your "Accept" recommendation.

Reviewer #1: (No Response)

Reviewer #2: (No Response)

2. Is the manuscript technically sound, and do the data support the conclusions?

Reviewer #1: Yes

Reviewer #2: Yes

3. Has the statistical analysis been performed appropriately and rigorously? 

Reviewer #1: Yes

Reviewer #2: Yes

4. Have the authors made all data underlying the findings in their manuscript fully available?

Reviewer #1: Yes

Reviewer #2: No

5. Is the manuscript presented in an intelligible fashion and written in standard English?

Reviewer #1: Yes

Reviewer #2: Yes

6. Review Comments to the Author

Reviewer #1: Dear Authors,

Thank you for your diligence in addressing comments from both reviewers. I hope you find the minor recommendations below helpful.

1. In the sample section you mention that your analytical sample is similar to the study's full sample in terms of age and gender distribution. It would be helpful to know if the samples are also similar in terms of parental education and income. If the data allow, please mention similarities and differences along other relevant dimensions of the data.

2. In the paragraph before the start of the "Statistic Analysis" section, you mention that the divorce group likely includes parents who split up after cohabiting. Please include some estimate from either national or regional data that gives the reader an idea as to what proportion of unions in this study were likely not marriages.

3. Your results table presents income measures in units of 1000NOK, while Table 1 describes income in units of 1NOK. You may want to present Table 1 income in units of 1000NOK as well to be consistent with results.

4. In the robustness checks you explore the correlation between parent's education and timing of divorce. You show an average difference of three years in the timing of divorce between the most and least educated parents. This translates into a difference between experiencing divorce around age 5 versus experiencing divorce around age 8. This difference may be important from a developmental framework. It would be worthwhile to control for these differences in your main regressions. Please add update your main models with a control measure for time since separation/divorce. The measure would equal zero for kids who did not experience parental divorce/separation, and it should equal time since event for kids who experienced divorce/separation. This should be a low cost update to models and figures.

Thank you for your work on this research.

Reviewer #2: I thank the authors for taking into account most of my comments and for further improving a already good study. However, I believe that they could conduct one robustness check. In response to my third point, the authors wrote that they do not have information on parental occupation and parental wealth. I see that they cannot then use these variables for a robustness check. However, they can use household income, a variable which is already present in their analysis as a control. I would like to see therefore at least a robustness check using household income as an alernative indicator of a family's socioeocnomic resources than parental education. It is precisely one shortocoming of the previous literature that they only used parental education and this study has the potential to address this shortcoming.

7. PLOS authors have the option to publish the peer review history of their article (what does this mean?). If published, this will include your full peer review and any attached files.

Reviewer #1: Yes: Ravaris L Moore

Reviewer #2: No

---

## [Author Response · Author response to Decision Letter 1]

30 Jan 2020

Author Response to Reviewers’ Comments

Journal: PLOS ONE

Manuscript ID: PONE-D-19-20535R1

Dear Tomáš Želinský, Ph.D.

Thank you again for allowing us to revise and resubmit our paper. Below are the comments from the reviewers (numbered) and our responses. We have uploaded one version of the manuscript where these changes are identified using track changes (Revised Manuscript with Track Changes”) and one version that is clean (Manuscript).

We thank you for your consideration of our revised manuscript and responses, and we look forward to hearing from you.

Sincerely,

Sondre Aa. Nilsen

(on behalf of all authors)

 

Response to Editorial comments:

1. Once again, I would like to thank you for your patience. Both referees were happy with your adjustments, and they only suggest minor revisions, which I believe, reflects the quality of the improvements you made. 

Both referees made a point related to the robustness checks, although from a different perspective. While Reviewer 1 would like to see whether adding "a control measure for time since separation/divorce" could affect your main findings, Reviewer 2 would like to see the results for an additional robustness check using household income as an alternative indicator of a family's socioeconomic resources. 

I believe that these suggestions will not affect the main results of your paper, and I will be happy to recommend your paper for acceptance. 

Response: Thank you for your availability and swift handling of this review process. We have made some additional changes based on the reviewer comments; 

- Detailed associations of “time since divorce” on the adolescents’ GPA

- across parental educational levels 

- Conducted additional robustness checks using household income

- We have also made some small changes based on reviewer 1 points 1-3, and fixed some typos and grammatical errors that we detected. 

The reviewers’ comments and our point-by-point responses can be found below. 

Response to Comments from Reviewer 1

We thank the reviewer for the efforts of evaluating our manuscript a second time, and for providing constructive means to further improve the paper. 

Reviewer comments:

1. In the sample section you mention that your analytical sample is similar to the study's full sample in terms of age and gender distribution. It would be helpful to know if the samples are also similar in terms of parental education and income. If the data allow, please mention similarities and differences along other relevant dimensions of the data.

Response: 

We agree that additional similarities/differences could be useful to add. Unfortunately, we only have register based information about parental income and education on the sample that consented to this register linkage (i.e. the sample that we investigate in this paper). We have, however, compared age, gender, and self-reported parental education and perceived economic well-being between the total sample and the sample consenting to the register linkage. We have uploaded this table as a supplementary file, as we think this information is best suited as a supplement to the paper. As shown in the table the samples were nearly identical across these dimensions between the two samples. 

We have updated the sample section which now reads;

All adolescents born between 1993 and 1995 and residing in Hordaland at the time of the survey were invited (N = 19,439) to participate, and 10,257 agreed, yielding a participation rate of 53 % for the entire study. The present paper is based on a subsample of 9,166 adolescents (47 % of the invited population) who consented to register linkage. This subsample was nearly identical to the total sample with regards to age and gender distribution, and self-reported sociodemographics (see S1 Table). 

2. In the paragraph before the start of the "Statistic Analysis" section, you mention that the divorce group likely includes parents who split up after cohabiting. Please include some estimate from either national or regional data that gives the reader an idea as to what proportion of unions in this study were likely not marriages.

Response: Thank you for this suggestion. We agree that additional information about marriage/cohabitation from regional data are useful to the paper. We have added the following based on available regional statistics to the paragraph just before the statistics section;

Official statistics from Hordaland county in 2012 report that 73.5 % of children and youth below the age of 18 in a two-parent household lived with married parents (the rest with cohabiting parents). Thus, the nondivorced group in the present study most likely contained a group of adolescents with parents that had cohabitated since their birth. As some cohabiting unions eventually marry, we find it likely that the proportion of cohabiting unions in the present sample was somewhat lower than regional estimates also including younger children. Similarly, the divorced group likely contained a group of adolescents whose parents split up from cohabitation. Unfortunately, no official statistics regarding dissolution from cohabiting unions in Norway exists. Our inability to exactly detail the adolescents’ family structure is not unique to the present study but has been rather common within this research field [2]

2. In the paragraph before the start of the "Statistic Analysis" section, you mention that the divorce group likely includes parents who split up after cohabiting. Please include some estimate from either national or regional data that gives the reader an idea as to what proportion of unions in this study were likely not marriages.

Response: Thank you for this suggestion. We agree that additional information about marriage/cohabitation from regional data are useful to the paper. We have added the following based on available regional statistics to the paragraph just before the statistics section;

Official statistics from Hordaland county in 2012 report that 73.5 % of children and youth below the age of 18 in a two-parent household lived with married parents (the rest with cohabiting parents). Thus, the nondivorced group in the present study most likely contained a group of adolescents with parents that had cohabitated since their birth. As some cohabiting unions eventually marry, we find it likely that the proportion of cohabiting unions in the present sample was somewhat lower than regional estimates also including younger children. Similarly, the divorced group likely contained a group of adolescents whose parents split up from cohabitation. Unfortunately, no official statistics regarding dissolution from cohabiting unions in Norway exists. Our inability to exactly detail the adolescents’ family structure is not unique to the present study but has been rather common within this research field [2]

3. Your results table presents income measures in units of 1000NOK, while Table 1 describes income in units of 1NOK. You may want to present Table 1 income in units of 1000NOK as well to be consistent with results.

Response: Thank you for noticing this inconsistency. We have updated Table 1 so that the units match. 

4. In the robustness checks you explore the correlation between parent's education and timing of divorce. You show an average difference of three years in the timing of divorce between the most and least educated parents. This translates into a difference between experiencing divorce around age 5 versus experiencing divorce around age 8. This difference may be important from a developmental framework. It would be worthwhile to control for these differences in your main regressions. Please add update your main models with a control measure for time since separation/divorce. The measure would equal zero for kids who did not experience parental divorce/separation, and it should equal time since event for kids who experienced divorce/separation. This should be a low cost update to models and figures.

Response: Thank you for this interesting suggestion. We have spent quite some time debating how adding a measure that equals zero for kids who do not divorce, and time since event for those who do divorce, may affect our models – and especially the interpretations of the models. Firstly, - we have checked this following your advice. The results when adjusting for this variable slightly strengthens all our main findings (i.e., makes the heterogenous effects slightly larger (by the second decimal point)), and do not change the interpretation of the significance level of any of the reported estimates. However, although this might be a mathematically sound method of adding an adjustment to a single level of a dichotomous categorical predictor in a regression (i.e., the “divorced-level” of the “divorce status” variable”), this also appear to permit calculations of impossible/paradoxical values based on the coefficients reported. For instance, the estimates of having non-divorced parents who divorced four years ago. 

We therefore prefer to not add this in our main reported analyses as it does not alter the main findings, and may induce some confusions to the interpretations of the estimates. We have, however, conducted further checks of the links between time since divorce, parental educational qualifications, and the adolescents’ GPA. We believe that adding this to the paper is perhaps more illuminating to the question of whether time since divorce is related to the adolescents’ GPA, and whether time since divorce potentially could influence our findings given the slight differences in time since divorce across parental educational qualifications. 

We have described this in detail in the statistics section;

The timing of divorce could potentially covary with parental educational levels and the adolescents’ GPA (e.g., if highly educated parents divorced later on, the estimates of divorce by parental education on the adolescents’ GPA could be influenced by the proximity to the event of dissolution). Moreover, it is possible that associations between timing of divorce and the adolescents’ GPA depend on parental educational qualifications (i.e., that more time spent with highly educated divorced parents differ to time spent with lowly educated divorced parents). We investigate these issues by comparing years since divorce across parental educational levels, and by graphically plotting potential linear and non-linear relationships between timing of divorce and GPA by parental educational qualifications. Generalized additive models (GAMs) were used to investigate potential non-linear relationships. In brief, GAMs may be considered as a semi-parametric extension of the generalized linear model, with the strength of the ability to detect non-linear structures in data that otherwise might be missed [52].

Added a paragraph to the results-section;

Adolescents with highly educated parents experienced, on average, that their parents divorced somewhat later (see Figs 3A and 3B). The mean difference in years since divorce among highly educated (i.e., ISCED 7-8) vs. lowly educated (i.e., ISCED 0-2) mothers was about 2.2 years, while the comparable figure among fathers was 2.8 years. Plotting the adolescents’ GPA as a function of years since divorce across the parental educational qualifications (see Fig 4) revealed a slight negative linear association between years since divorce and GPA across most of both maternal and paternal educational levels. The negative association between time since divorce and GPA was strongest among lowly educated mothers. As lowly educated mothers on average had most years since divorce, this finding highlights that time since divorce could not explain the heterogeneity in the associations between divorce and the adolescents’ GPA by maternal educational qualifications. Indeed, the plot suggests that holding years since divorce constant across maternal educational qualifications would slightly strengthen the difference in the negative association between divorce and GPA among adolescents with highly- compared to lowly educated mothers. 

The plotted GAM curves show some variability around the linear functions for some of the parental educational levels. Overall, these trends do not give any strong indications that GPA is highly influenced by the timing of divorce in the present study.

And added a paragraph to the discussion;

Adolescents with highly educated parents experienced, on average, that their parents had divorced somewhat more recently. Time since divorce was overall weakly and negatively associated with the adolescents’ GPA, with some small observable variations across parental educational qualifications. These variations could, however, not explain the heterogeneity found in the present study. Overall, this is generally in keeping with studies that find that youth’s post-divorce adjustment is quite stable or gradually slightly worsen as time passes since the divorce [61,62]. 

Response to Comments from Reviewer 2

1. Reviewer #2: I thank the authors for taking into account most of my comments and for further improving a already good study. However, I believe that they could conduct one robustness check. In response to my third point, the authors wrote that they do not have information on parental occupation and parental wealth. I see that they cannot then use these variables for a robustness check. However, they can use household income, a variable which is already present in their analysis as a control. I would like to see therefore at least a robustness check using household income as an alernative indicator of a family's socioeocnomic resources than parental education. It is precisely one shortocoming of the previous literature that they only used parental education and this study has the potential to address this shortcoming.

Response: We thank the reviewer for assessing our paper a second time, and for the positive comments and encouragements to further improve the manuscript 

We have followed the reviewer’s advice and added additional checks utilizing household income as an alternative indicator of socioeconomic status. Utilizing the equivalized disposable income measure divided into quartiles, we find a similar but weaker pattern than in the models with parental education. We have added a figure summarizing the regression results and uploaded it as a supplementary figure. 

We have updated the statistics section; 

Lastly, we performed checks utilizing the income measures as alternative indicators of the family’s socioeconomic resources. The income measures were divided into quartiles (i.e., into four equal parts representing the lowest 25 % to the highest 25%), and the adolescents’ GPA was regressed on the interaction term between parental divorce and the income quartiles (similarly to the procedure described above).

And made changes to the result section;

Lastly, using equivalized disposable income (EDI) as an alternative indicator of socioeconomic resources, we found a similar but weaker pattern whereby the negative association between divorce and GPA was relatively stronger among adolescents in the second income quartile (Q2; b = - 0.16, p < 0.01) and in the fourth quartile (Q4; b = -0.15, p = 0.02) compared to those in the first quartile (Q1). The difference between Q1 and Q3 was not statistically significant (see S1 Fig. with further test statistics). Adjusting the analyses for parental educational qualifications attenuated and removed the significant difference between Q1 and Q4. No heterogeneity in the associations between divorce and GPA by mothers’ or fathers’ net income were found (results not shown). 

Of note, it is highly likely that potential heterogeneity by measures of household income in the links between divorce and academic outcomes is sensitive to how income is operationalized. As parental education was the main focus of interest in the present study, we did not examine this any further in the present paper (e.g., other ways of dividing income into categories).

---

## [Editor Report · Decision Letter 2]

3 Feb 2020

Divorce and adolescent academic achievement: Heterogeneity in the associations by parental education

PONE-D-19-20535R2

Dear Dr. Nilsen,

We are pleased to inform you that your manuscript has been judged scientifically suitable for publication and will be formally accepted for publication once it complies with all outstanding technical requirements.

With kind regards,

Tomáš Želinský, Ph.D.

Academic Editor

PLOS ONE

Additional Editor Comments (optional):

Once again, I would like to thank you for addressing reviewers' comments, which, I believe, contributed to improving the quality of your paper. 
---

## [Editor Report · Acceptance letter]

7 Feb 2020

PONE-D-19-20535R2 

Divorce and adolescent academic achievement: Heterogeneity in the associations by parental education 

Dear Dr. Nilsen:

I am pleased to inform you that your manuscript has been deemed suitable for publication in PLOS ONE. Congratulations! Your manuscript is now with our production department. 

With kind regards,

on behalf of

Dr. Tomáš Želinský 

Academic Editor

PLOS ONE